# Flow Matching for Scalable Simulation-Based Inference

**Jonas Wildberger***
Max Planck Institute for Intelligent Systems
Tübingen, Germany
`wildberger.jonas@tuebingen.mpg.de`

**Maximilian Dax***
Max Planck Institute for Intelligent Systems
Tübingen, Germany
`maximilian.dax@tuebingen.mpg.de`

**Simon Buchholz***
Max Planck Institute for Intelligent Systems
Tübingen, Germany
`sbuchholz@tue.mpg.de`

**Stephen R. Green**
University of Nottingham
Nottingham, United Kingdom

**Jakob H. Macke**
Max Planck Institute for Intelligent Systems &
Machine Learning in Science,
University of Tübingen
Tübingen, Germany

**Bernhard Schölkopf**
Max Planck Institute for Intelligent Systems
Tübingen, Germany

## Abstract

Neural posterior estimation methods based on discrete normalizing flows have become established tools for simulation-based inference (SBI), but scaling them to high-dimensional problems can be challenging. Building on recent advances in generative modeling, we here present flow matching posterior estimation (FMPE), a technique for SBI using continuous normalizing flows. Like diffusion models, and in contrast to discrete flows, flow matching allows for unconstrained architectures, providing enhanced flexibility for complex data modalities. Flow matching, therefore, enables exact density evaluation, fast training, and seamless scalability to large architectures—making it ideal for SBI. We show that FMPE achieves competitive performance on an established SBI benchmark, and then demonstrate its improved scalability on a challenging scientific problem: for gravitational-wave inference, FMPE outperforms methods based on comparable discrete flows, reducing training time by 30% with substantially improved accuracy. Our work underscores the potential of FMPE to enhance performance in challenging inference scenarios, thereby paving the way for more advanced applications to scientific problems.

## 1 Introduction

The ability to readily represent Bayesian posteriors of arbitrary complexity using neural networks would herald a revolution in scientific data analysis. Such networks could be trained using simulated data and used for amortized inference across observations—bringing tractable inference and speed to a myriad of scientific models. Thanks to innovative architectures such as normalizing flows [1, 2], approaches to neural simulation-based inference (SBI) [3] have seen remarkable progress in recent years. Here, we show that modern approaches to deep generative modeling (particularly flow matching) deliver substantial improvements in simplicity, flexibility and scaling when adapted to SBI.

---

*Equal contribution

37th Conference on Neural Information Processing Systems (NeurIPS 2023).

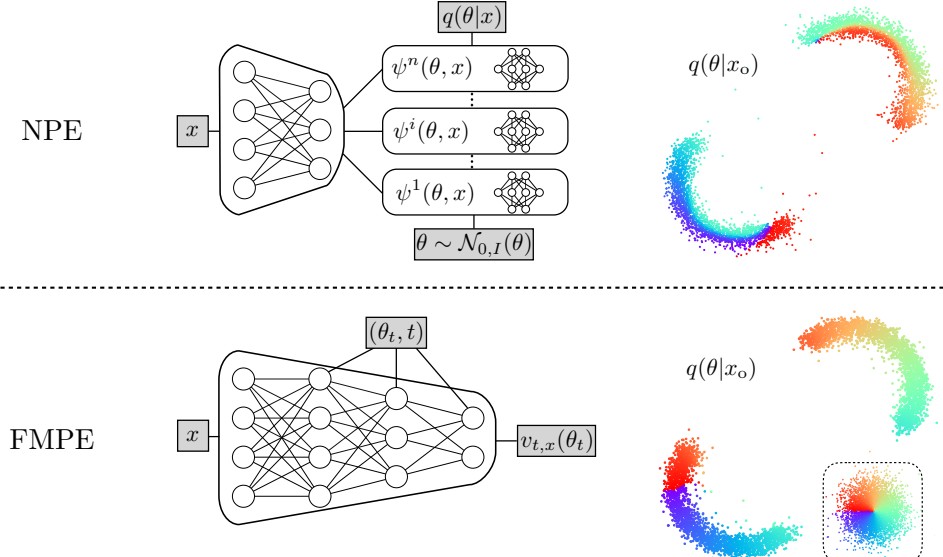

Figure 1: Comparison of network architectures (left) and flow trajectories (right). Discrete flows (NPE, top) require a specialized architecture for the density estimator. Continuous flows (FMPE, bottom) are based on a vector field parametrized with an unconstrained architecture. FMPE uses this additional flexibility to put an enhanced emphasis on the conditioning data $x$, which in the SBI context is typically high dimensional and in a complex domain. Further, the optimal transport path produces simple flow trajectories from the base distribution (inset) to the target.

The Bayesian approach to data analysis is to compare observations to models via the posterior distribution $p(\theta|x)$. This gives our degree of belief that model parameters $\theta$ gave rise to an observation $x$, and is proportional to the model likelihood $p(x|\theta)$ times the prior $p(\theta)$. One is typically interested in representing the posterior in terms of a collection of samples, however obtaining these through standard likelihood-based algorithms can be challenging for intractable or expensive likelihoods. In such cases, SBI offers an alternative based instead on *data simulations* $x \sim p(x|\theta)$. Combined with deep generative modeling, SBI becomes a powerful paradigm for scientific inference [3]. Neural posterior estimation (NPE) [4–6], for instance, trains a conditional density estimator $q(\theta|x)$ to approximate the posterior, allowing for rapid sampling and density estimation for any $x$ consistent with the training distribution.

The NPE density estimator $q(\theta|x)$ is commonly taken to be a (discrete) normalizing flow [1, 2], an approach that has brought state-of-the-art performance in challenging problems such as gravitational-wave inference [7]. Naturally, performance hinges on the expressiveness of $q(\theta|x)$. Normalizing flows transform noise to samples through a discrete sequence of basic transforms. These have been carefully engineered to be invertible with simple Jacobian determinant, enabling efficient maximum likelihood training, while at the same time producing expressive $q(\theta|x)$. Although many such discrete flows are universal density approximators [2], in practice, they can be challenging to scale to very large networks, which are needed for big-data experiments.

Recent studies [8, 9] propose neural posterior score estimation (NPSE), a rather different approach that models the posterior distribution with score-matching (or diffusion) networks. These techniques were originally developed for generative modeling [10–12], achieving state-of-the-art results in many domains, including image generation [13, 14]. Like discrete normalizing flows, diffusion models transform noise into samples, but with trajectories parametrized by a *continuous* "time" parameter $t$. The trajectories solve a stochastic differential equation [15] (SDE) defined in terms of a vector field $v_t$, which is trained to match the score of the intermediate distributions $p_t$. NPSE has several advantages compared to NPE, including the ability to combine multiple observations at inference time [9] and, importantly, the freedom to use unconstrained network architectures.

We here propose to use flow matching, another recent technique for generative modeling, for Bayesian inference, an approach we refer to as flow-matching posterior estimation (FMPE). Flow matching is also based on a vector field $v_t$ and thereby also admits flexible network architectures (Fig. 1). For flow

matching, however, $v_t$ directly defines the velocity field of sample trajectories, which solve ordinary differential equations (ODEs) and are deterministic. As a consequence, flow matching allows for additional freedom in designing non-diffusion paths such as optimal transport, and provides direct access to the density [16]. These differences are summarized in Tab. 1.

|  | NPE | NPSE | **FMPE (Ours)** |
|---|---|---|---|
| Tractable posterior density | Yes | No | Yes |
| Unconstrained network architecture | No | Yes | Yes |
| Network passes for sampling | Single | Many | Many |

Table 1: Comparison of posterior-estimation methods.

Our contributions are as follows:

- We adapt flow-matching to Bayesian inference, proposing FMPE. In general, the modeling requirements of SBI are different from generative modeling. For the latter, sample quality is critical, i.e., that samples lie in the support of a complex distribution (e.g., images). In contrast, for SBI, $p(\theta|x)$ is typically less complex for fixed $x$, but $x$ itself can be complex and high-dimensional. We therefore consider pyramid-like architectures from $x$ to $v_t$, with gated linear units to incorporate $(\theta, t)$ dependence, rather than the typical U-Net used for images (Fig. 1). We also propose an alternative $t$-weighting in the loss, which improves performance in many tasks.

- Under certain regularity assumptions, we prove an upper bound on the KL divergence between the model and target posterior. This implies that estimated posteriors are mass-covering, i.e., that their support includes all $\theta$ consistent with observed $x$, which is highly desirable for scientific applications [17].

- We perform a number of experiments to investigate the performance of FMPE.[2] Our two-pronged approach, which involves a set of benchmark tests and a real-world problem, is designed to probe complementary aspects of the method, covering breadth and depth of applications. First, on an established suite of SBI benchmarks, we show that FMPE performs comparably—or better—than NPE across most tasks, and in particular exhibits mass-covering posteriors in all cases (Sec. 4). We then push the performance limits of FMPE on a challenging real-world problem by turning to gravitational-wave inference (Sec. 5). We show that FMPE substantially outperforms an NPE baseline in terms of training time, posterior accuracy, and scaling to larger networks.

## 2 Preliminaries

**Normalizing flows.** A normalizing flow [1, 2] defines a probability distribution $q(\theta|x)$ over parameters $\theta \in \mathbb{R}^n$ in terms of an invertible mapping $\psi_x : \mathbb{R}^n \to \mathbb{R}^n$ from a simple base distribution $q_0(\theta)$,

$$q(\theta|x) = (\psi_x)_* q_0(\theta) = q_0(\psi_x^{-1}(\theta)) \det \left| \frac{\partial \psi_x^{-1}(\theta)}{\partial \theta} \right|, \tag{1}$$

where $(\cdot)_*$ denotes the pushforward operator, and for generality we have conditioned on additional context $x \in \mathbb{R}^m$. Unless otherwise specified, a normalizing flow refers to a *discrete* flow, where $\psi_x$ is given by a composition of simpler mappings with triangular Jacobians, interspersed with shuffling of the $\theta$. This construction results in expressive $q(\theta|x)$ and also efficient density evaluation [2].

**Continuous normalizing flows.** A continuous flow [18] also maps from base to target distribution, but is parametrized by a continuous "time" $t \in [0, 1]$, where $q_0(\theta|x) = q_0(\theta)$ and $q_1(\theta|x) = q(\theta|x)$. For each $t$, the flow is defined by a vector field $v_{t,x}$ on the sample space.[3] This corresponds to the velocity of the sample trajectories,

$$\frac{d}{dt} \psi_{t,x}(\theta) = v_{t,x}(\psi_{t,x}(\theta)), \qquad \psi_{0,x}(\theta) = \theta. \tag{2}$$

---

[2]Code available here.

[3]In the SBI literature, this is also commonly referred to as "parameter space".

We obtain the trajectories $\theta_t \equiv \psi_{t,x}(\theta)$ by integrating this ODE. The final density is given by

$$q(\theta|x) = (\psi_{1,x})_* q_0(\theta) = q_0(\theta) \exp\left(-\int_0^1 \operatorname{div} v_{t,x}(\theta_t)\, \mathrm{d}t\right), \tag{3}$$

which is obtained by solving the transport equation $\partial_t q_t + \operatorname{div}(q_t v_{t,x}) = 0$.

The advantage of the continuous flow is that $v_{t,x}(\theta)$ can be simply specified by a neural network taking $\mathbb{R}^{n+m+1} \to \mathbb{R}^n$, in which case (2) is referred to as a *neural ODE* [18]. Since the density is tractable via (3), it is in principle possible to train the flow by maximizing the (log-)likelihood. However, this is often not feasible in practice, since both sampling and density estimation require many network passes to numerically solve the ODE (2).

**Flow matching.**   An alternative training objective for continuous normalizing flows is provided by flow matching [16]. This directly regresses $v_{t,x}$ on a vector field $u_{t,x}$ that generates a target probability path $p_{t,x}$. It has the advantage that training does not require integration of ODEs, however it is not immediately clear how to choose $(u_{t,x}, p_{t,x})$. The key insight of [16] is that, if the path is chosen on a *sample-conditional* basis,[4] then the training objective becomes extremely simple. Indeed, given a sample-conditional probability path $p_t(\theta|\theta_1)$ and a corresponding vector field $u_t(\theta|\theta_1)$, we specify the sample-conditional flow matching loss as

$$\mathcal{L}_{\text{SCFM}} = \mathbb{E}_{t\sim\mathcal{U}[0,1],\, x\sim p(x),\, \theta_1\sim p(\theta|x),\, \theta_t\sim p_t(\theta_t|\theta_1)} \left\| v_{t,x}(\theta_t) - u_t(\theta_t|\theta_1) \right\|^2. \tag{4}$$

Remarkably, minimization of this loss is equivalent to regressing $v_{t,x}(\theta)$ on the *marginal* vector field $u_{t,x}(\theta)$ that generates $p_t(\theta|x)$ [16]. Note that in this expression, the $x$-dependence of $v_{t,x}(\theta)$ is picked up via the expectation value, with the sample-conditional vector field independent of $x$.

There exists considerable freedom in choosing a sample-conditional path. Ref. [16] introduces the family of Gaussian paths

$$p_t(\theta|\theta_1) = \mathcal{N}(\theta|\mu_t(\theta_1), \sigma_t(\theta_1)^2 I_n), \tag{5}$$

where the time-dependent means $\mu_t(\theta_1)$ and standard deviations $\sigma_t(\theta_1)$ can be freely specified (subject to boundary conditions[5]). For our experiments, we focus on the optimal transport paths defined by $\mu_t(\theta_1) = t\theta_1$ and $\sigma_t(\theta_1) = 1 - (1 - \sigma_{\min})t$ (also introduced in [16]). The sample-conditional vector field then has the simple form

$$u_t(\theta|\theta_1) = \frac{\theta_1 - (1 - \sigma_{\min})\theta}{1 - (1 - \sigma_{\min})t}. \tag{6}$$

**Neural posterior estimation (NPE).**   NPE is an SBI method that directly fits a density estimator $q(\theta|x)$ (usually a normalizing flow) to the posterior $p(\theta|x)$ [4–6]. NPE trains with the maximum likelihood objective $\mathcal{L}_{\text{NPE}} = -\mathbb{E}_{p(\theta)p(x|\theta)} \log q(\theta|x)$, using Bayes' theorem to simplify the expectation value with $\mathbb{E}_{p(x)p(\theta|x)} \to \mathbb{E}_{p(\theta)p(x|\theta)}$. During training, $\mathcal{L}_{\text{NPE}}$ is estimated based on an empirical distribution consisting of samples $(\theta, x) \sim p(\theta)p(x|\theta)$. Once trained, NPE can perform inference for every new observation using $q(\theta|x)$, thereby *amortizing* the computational cost of simulation and training across all observations. NPE further provides exact density evaluations of $q(\theta|x)$. Both of these properties are crucial for the physics application in section 5, so we aim to retain these properties with FMPE.

**Related work**

Flow matching [16] has been developed as a technique for generative modeling, and similar techniques are discussed in [19–21] and extended in [22, 23]. Flow matching encompasses the deterministic ODE version of diffusion models [10–12] as a special instance. Although to our knowledge flow matching has not previously been applied to Bayesian inference, score-matching diffusion models have been proposed for SBI in [8, 9] with impressive results. These studies, however, use stochastic formulations via SDEs [15] or Langevin steps and are therefore not directly applicable when evaluations of the posterior density are desired (see Tab. 1). It should be noted that score modeling can also be used

---

[4]We refer to conditioning on $\theta_1$ as *sample*-conditioning to distinguish from conditioning on $x$.

[5]The sample-conditional probability path should be chosen to be concentrated around $\theta_1$ at $t = 1$ (within a small region of size $\sigma_{\min}$) and to be the base distribution at $t = 0$.

to parameterize continuous normalizing flows via an ODE. Extension of [8, 9] to the deterministic formulation could thereby be seen as a special case of flow matching. Many of our analyses and the practical guidance provided in Section 3 therefore also apply to score matching.

We here focus on comparisons of FMPE against NPE [4–6], as it best matches the requirements of the application in section 5. Other SBI methods include approximate Bayesian computation [24–28], neural likelihood estimation [29–32] and neural ratio estimation [33–39]. Many of these approaches have sequential versions, where the estimator networks are specifically tuned to a specific observation $x_\text{o}$. FMPE has a tractable density, so it is straightforward to apply the sequential NPE [4–6] approaches to FMPE. In this case, inference is no longer amortized, so we leave this extension to future work.

## 3 Flow matching posterior estimation

To apply flow matching to SBI we use Bayes' theorem to make the usual replacement $\mathbb{E}_{p(x)p(\theta|x)} \to \mathbb{E}_{p(\theta)p(x|\theta)}$ in the loss function (4), eliminating the intractable expectation values. This gives the FMPE loss

$$\mathcal{L}_{\text{FMPE}} = \mathbb{E}_{t\sim p(t), \theta_1 \sim p(\theta), x\sim p(x|\theta_1), \theta_t \sim p_t(\theta_t|\theta_1)} \left\| v_{t,x}(\theta_t) - u_t(\theta_t|\theta_1) \right\|^2, \tag{7}$$

which we minimize using empirical risk minimization over samples $(\theta, x) \sim p(\theta)p(x|\theta)$. In other words, training data is generated by sampling $\theta$ from the prior, and then simulating data $x$ corresponding to $\theta$. This is in close analogy to NPE training, but replaces the log likelihood maximization with the sample-conditional flow matching objective. Note that in this expression we also sample $t \sim p(t)$, $t \in [0, 1]$ (see Sec. 3.3), which generalizes the uniform distribution in (4). This provides additional freedom to improve learning in our experiments.

### 3.1 Probability mass coverage

As we show in our examples, trained FMPE models $q(\theta|x)$ can achieve excellent results in approximating the true posterior $p(\theta|x)$. However, it is not generally possible to achieve *exact* agreement due to limitations in training budget and network capacity. It is therefore important to understand how inaccuracies manifest. Whereas sample quality is the main criterion for generative modeling, for scientific applications one is often interested in the overall shape of the distribution. In particular, an important question is whether $q(\theta|x)$ is *mass-covering*, i.e., whether it contains the full support of $p(\theta|x)$. This minimizes the risk to falsely rule out possible explanations of the data. It also allows us to use importance sampling if the likelihood $p(x|\theta)$ of the forward model can be evaluated, which can be used for precise estimation of the posterior [40, 41].

Consider first the mass-covering property for NPE. NPE directly minimizes the forward KL divergence $D_{\text{KL}}(p(\theta|x)||q(\theta|x))$, and thereby provides probability-mass covering results. Therefore, even if NPE is not accurately trained, the estimate $q(\theta|x)$ should cover the entire support of the posterior $p(\theta|x)$ and the failure to do so can be observed in the validation loss. As an illustration in an unconditional setting, we observe that a unimodal Gaussian $q$ fitted to a bimodal target distribution $p$ captures both modes when using the forward KL divergence $D_{\text{KL}}(p||q)$, but only a single mode when using the backwards direction $D_{\text{KL}}(q||p)$ (Fig. 2).

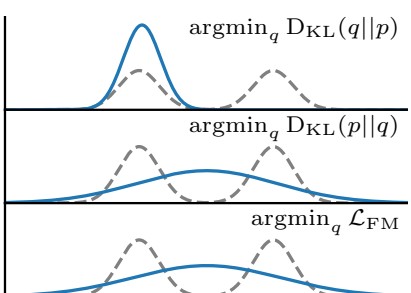

Figure 2: A Gaussian (blue) fitted to a bimodal distribution (gray) with various objectives.

For FMPE, we can fit a Gaussian flow-matching model $q(\theta) = \mathcal{N}(\hat{\mu}, \hat{\sigma}^2)$ to the same bimodal target, in this case, parametrizing the vector field as

$$v_t(\theta) = \frac{(\sigma_t^2 + (t\hat{\sigma})^2 - \sigma_t)\theta_t + t\hat{\mu} \cdot \sigma_t}{t \cdot (\sigma_t^2 + (t\hat{\sigma})^2)} \tag{8}$$

(see Appendix A), we also obtain a mass-covering distribution when fitting the learnable parameters $(\hat{\mu}, \hat{\sigma})$ via (4). This provides some indication that the flow matching objective induces mass-covering behavior, and leads us to investigate the more general question of whether the mean squared error

between vector fields $u_t$ and $v_t$ bounds the forward KL divergence. Indeed, the former agrees up to constant with the sample-conditional loss (4) (see Sec. 2).

We denote the flows of $u_t, v_t$, by $\phi_t, \psi_t$, respectively, and we set $q_t = (\psi_t)_* q_0$, $p_t = (\phi_t)_* q_0$. The precise question then is whether we can bound $\mathrm{D}_{\mathrm{KL}}(p_1 || q_1)$ by $\mathrm{MSE}_p(u, v)^\alpha$ for some positive power $\alpha$. It was already observed in [42] that this is not true in general, and we provide a simple example to that effect in Lemma 1 in Appendix B. Indeed, it was found in [42] that to bound the forward KL divergence we also need to control the Fisher divergence, $\int p_t(\mathrm{d}\theta)(\nabla \ln p_t(\theta) - \nabla q_t(\theta))^2$.

Here we show instead that a bound can be obtained under sufficiently strong regularity assumptions on $p_0$, $u_t$, and $v_t$. The following statement is slightly informal, and we refer to the supplement for the complete version.

**Theorem 1.** *Let $p_0 = q_0$ and assume $u_t$ and $v_t$ are two vector fields whose flows satisfy $p_1 = (\phi_1)_* p_0$ and $q_1 = (\psi_1)_* q_0$. Assume that $p_0$ is square integrable and satisfies $|\nabla \ln p_0(\theta)| \leq c(1 + |\theta|)$ and $u_t$ and $v_t$ have bounded second derivatives. Then there is a constant $C > 0$ such that (for $\mathrm{MSE}_p(u, v) < 1$)*

$$\mathrm{D}_{\mathrm{KL}}(p_1 || q_1) \leq C \, \mathrm{MSE}_p(u, v)^{\frac{1}{2}}. \tag{9}$$

*The proof of this result can be found in appendix B.* While the regularity assumptions are not guaranteed to hold in practice when $v_t$ is parametrized by a neural net, the theorem nevertheless gives some indication that the flow-matching objective encourages mass coverage. In Section 4 and 5, this is complemented with extensive empirical evidence that flow matching indeed provides mass-covering estimates.

We remark that it was shown in [43] that the KL divergence of SDE solutions can be bounded by the MSE of the estimated score function. Thus, the smoothing effect of the noise ensures mass coverage, an aspect that was further studied using the Fokker-Planck equation in [42]. For flow matching, imposing the regularity assumption plays a similar role.

### 3.2 Network architecture

Generative diffusion or flow matching models typically operate on complicated and high dimensional data in the $\theta$ space (e.g., images with millions of pixels). One typically uses U-Net [44] like architectures, as they provide a natural mapping from $\theta$ to a vector field $v(\theta)$ of the same dimension. The dependence on $t$ and an (optional) conditioning vector $x$ is then added on top of this architecture.

For SBI, the data $x$ is often associated with a complicated domain, such as image or time series data, whereas parameters $\theta$ are typically low dimensional. In this context, it is therefore useful to build the architecture starting as a mapping from $x$ to $v(x)$ and then add conditioning on $\theta$ and $t$. In practice, one can therefore use any established feature extraction architecture for data in the domain of $x$, and adjust the dimension of the feature vector to $n = \dim(\theta)$. In our experiments, we found that the $(t, \theta)$-conditioning is best achieved using gated linear units [45] to the hidden layers of the network (see also Fig. 1); these are also commonly used for conditioning discrete flows on $x$.

### 3.3 Re-scaling the time prior

The time prior $\mathcal{U}[0, 1]$ in (4) distributes the training capacity uniformly across $t$. We observed that this is not always optimal in practice, as the complexity of the vector field may depend on $t$. For FMPE we therefore sample $t$ in (7) from a power-law distribution $p_\alpha(t) \propto t^{1/(1+\alpha)}$, $t \in [0, 1]$, introducing an additional hyperparameter $\alpha$. This includes the uniform distribution for $\alpha = 0$, but for $\alpha > 0$, assigns greater importance to the vector field for larger values of $t$. We empirically found this to improve learning for distributions with sharp bounds (e.g., Two Moons in Section 4).

## 4 SBI benchmark

We now evaluate FMPE on ten tasks included in the benchmark presented in [46], ranging from simple Gaussian toy models to more challenging SBI problems from epidemiology and ecology, with varying dimensions for parameters ($\dim(\theta) \in [2, 10]$) and observations ($\dim(x) \in [2, 100]$). For each task, we train three separate FMPE models with simulation budgets $N \in \{10^3, 10^4, 10^5\}$. We

use a simple network architecture consisting of fully connected residual blocks [47] to parameterize the conditional vector field. For the two tasks with $\dim(x) = 100$ (B-GLM-Raw, SLCP-D), we condition on $(t, \theta)$ via gated linear units as described in Section 3.2 (Fig. 8 in Appendix C shows the corresponding performance gain). For the remaining tasks with $\dim(x) \leq 10$ we concatenate $(t, \theta, x)$ instead. We reserve 5% of the simulations for validation. See Appendix C for details.

For each task and simulation budget, we evaluate the model with the lowest validation loss by comparing $q(\theta|x)$ to the reference posteriors $p(\theta|x)$ provided in [46] for ten different observations $x$ in terms of the C2ST score [48, 49]. This performance metric is computed by training a classifier to discriminate inferred samples $\theta \sim q(\theta|x)$ from reference samples $\theta \sim p(\theta|x)$. The C2ST score is then the test accuracy of this classifier, ranging from 0.5 (best) to 1.0. We observe that FMPE exhibits comparable performance to an NPE baseline model for most tasks and outperforms on several (Fig. 4). In terms of the MMD metric (Fig. 6 in the Appendix), FMPE clearly outperforms NPE (but MMD can be sensitive to its hyperparameters [46]). As NPE is one of the highest ranking methods for many tasks in the benchmark, these results show that FMPE indeed performs competitively with other existing SBI methods. We report an additional baseline for score matching in Fig. 7 in the Appendix.

As NPE and FMPE both directly target the posterior with a density estimator (in contrast to most other SBI methods), observed differences can be primarily attributed to their different approaches for density estimation. Interestingly, a great performance improvement of FMPE over NPE is observed for SLCP with a large simulation budget ($N = 10^5$). The SLCP task is specifically designed to have a simple likelihood but a complex posterior, and the FMPE performance underscores the enhanced flexibility of the FMPE density estimator.

Finally, we empirically investigate the mass coverage suggested by our theoretical analysis in Section 3.1. We display the density $\log q(\theta|x)$ of the reference samples $\theta \sim p(\theta|x)$ under our FMPE model $q$ as a histogram (Fig. 3). All samples $\theta \sim p(\theta|x)$ fall into the support from $q(\theta|x)$. This becomes apparent when comparing to the density $\log q(\theta|x)$ for samples $\theta \sim q(\theta|x)$ from $q$ itself. This FMPE result is therefore mass covering. Note that this does not necessarily imply conservative posteriors (which is also not generally true for the forward KL divergence [17, 50, 51]), and some parts of $p(\theta|x)$ may still be undersampled. Probability mass coverage, however, implies that no part is entirely missed (compare Fig. 2), even for multimodal distributions such as Two Moons. Fig. 9 in the Appendix confirms the mass coverage for the other benchmark tasks.

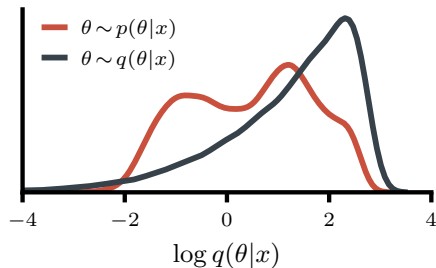

Figure 3: Histogram of FMPE densities $\log q(\theta|x)$ for reference samples $\theta \sim p(\theta|x)$ (Two Moons task, $N = 10^3$). The estimate $q(\theta|x)$ clearly covers $p(\theta|x)$ entirely.

## 5 Gravitational-wave inference

### 5.1 Background

Gravitational waves (GWs) are ripples of spacetime predicted by Einstein and produced by cataclysmic cosmic events such as the mergers of binary black holes (BBHs). GWs propagate across the universe to Earth, where the LIGO-Virgo-KAGRA observatories measure faint time-series signals embedded in noise. To-date, roughly 90 detections of merging black holes and neutron stars have been made [52], all of which have been characterized using Bayesian inference to compare against theoretical models.[6] These have yielded insights into the origin and evolution of black holes [53], fundamental properties of matter and gravity [54, 55], and even the expansion rate of the universe [56]. Under reasonable assumptions on detector noise, the GW likelihood is tractable,[7] and inference is

---

[6]BBH parameters $\theta \in \mathbb{R}^{15}$ include black-hole masses, spins, and the spacetime location and orientation of the system (see Tab. 4 in the Appendix). We represent $x$ in frequency domain; for two LIGO detectors and complex $f \in [20, 512]$ Hz, $\Delta f = 0.125$ Hz, we have $x \in \mathbb{R}^{15744}$.

[7]Noise is assumed to be stationary and Gaussian, so for frequency-domain data, the GW likelihood $p(x|\theta) = \mathcal{N}(h(\theta)|S_n)(x)$. Here $h(\theta)$ is a theoretical signal model based on Einstein's theory of general relativity, and $S_n$ is the power spectral density of the detector noise.

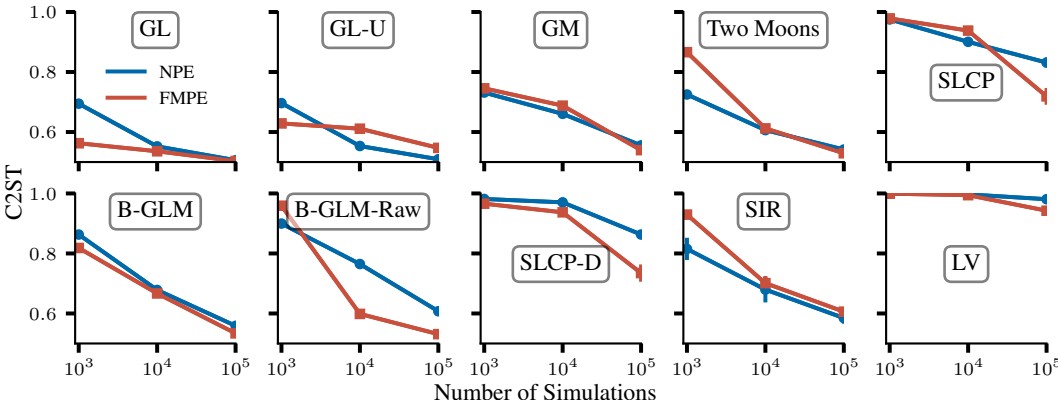

Figure 4: Comparison of FMPE with NPE, a standard SBI method, across 10 benchmark tasks [46].

typically performed using tools [57–60] based on Markov chain Monte Carlo [61, 62] or nested sampling [63] algorithms. This can take from hours to months, depending on the nature of the event and the complexity of the signal model, with a typical analysis requiring up to $\sim 10^8$ likelihood evaluations. The ever-increasing rate of detections means that these analysis times risk becoming a bottleneck. SBI offers a promising solution for this challenge that has thus been actively studied in the literature [64–68, 7, 69, 70, 41]. A fully amortized NPE-based method called DINGO recently achieved accuracies comparable to stochastic samplers with inference times of less than a minute per event [7]. To achieve accurate results, however, DINGO uses group-equivariant NPE [7, 69] (GNPE), an NPE extension that integrates known conditional symmetries. GNPE, therefore, does not provide a tractable density, which is problematic when verifying and correcting inference results using importance sampling [41].

## 5.2 Experiments

We here apply FMPE to GW inference. As a baseline, we train an NPE network with the settings described in [7] with a few minor changes (see Appendix D).[8] This uses an embedding network [71] to compress $x$ to a 128-dimensional feature vector, which is then used to condition a neural spline flow [72]. The embedding network consists of a learnable linear layer initialized with principal components of GW simulations followed by a series of dense residual blocks [47]. This architecture is a powerful feature extractor for GW measurements [7]. As pointed out in Section 3.2, it is straightforward to reuse such architectures for FMPE, with the following three modifications: (1) we provide the conditioning on $(t, \theta)$ to the network via gated linear units in each hidden layer; (2) we change the dimension of the final feature vector to the dimension of $\theta$ so that the network parameterizes the conditional vector field $(t, x, \theta) \rightarrow v_{t,x}(\theta)$; (3) we increase the number and width of the hidden layers to use the capacity freed up by removing the discrete normalizing flow.

We train the NPE and FMPE networks with $5 \cdot 10^6$ simulations for 400 epochs using a batch size of 4096 on an A100 GPU. The FMPE network ($1.9 \cdot 10^8$ learnable parameters, training takes $\approx 2$ days) is larger than the NPE network ($1.3 \cdot 10^8$ learnable parameters, training takes $\approx 3$ days), but trains substantially faster. We evaluate both networks on GW150914 [73], the first detected GW. We generate a reference posterior using the method described in [41]. Fig. 5 compares the inferred posterior distributions qualitatively and quantitatively in terms of the Jensen-Shannon divergence (JSD) to the reference.[9]

FMPE substantially outperforms NPE in terms of accuracy, with a mean JSD of $0.5$ mnat (NPE: $3.6$ mnat), and max JSD $< 2.0$ mnat, an indistinguishability criterion for GW posteriors [59]. Remarkably, FMPE accuracy is even comparable to GNPE, which leverages physical symmetries

---

[8]Our implementation builds on the public DINGO code from `https://github.com/dingo-gw/dingo`.

[9]We omit the three parameters $\phi_c, \phi_{JL}, \theta_{JN}$ in the evaluation as we use phase marginalization in importance sampling and the reference therefore uses a different basis for these parameters [41]. For GNPE we report the results from [7], which are generated with slightly different data conditioning. Therefore, we do not display the GNPE results in the corner plot, and the JSDs serve only as a rough comparison. The JSD for the $t_c$ parameter is not reported in [7] due to a $t_c$ marginalized reference.

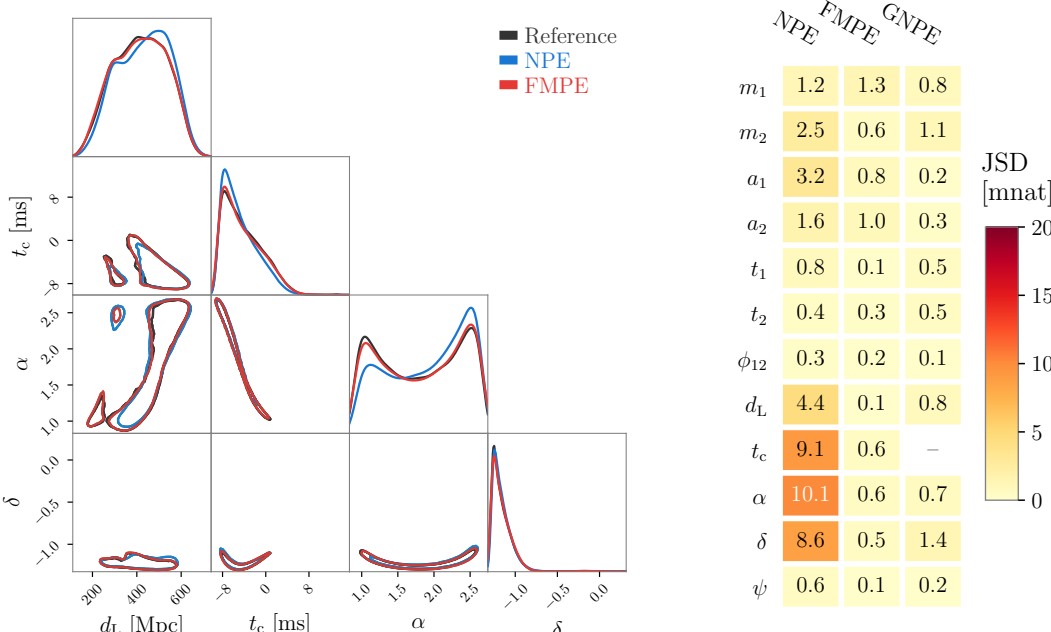

Figure 5: Results for GW150914 [73]. Left: Corner plot showing 1D marginals on the diagonal and 2D 50% credible regions. We display four GW parameters (distance $d_{\rm L}$, time of arrival $t_{\rm c}$, and sky coordinates $\alpha, \delta$); these represent the least accurate NPE parameters. Right: Deviation between inferred posteriors and the reference, quantified by the Jensen-Shannon divergence (JSD). The FMPE posterior matches the reference more accurately than NPE, and performs similarly to symmetry-enhanced GNPE. (We do not display GNPE results on the left due to different data conditioning settings in available networks.)

to simplify data. Finally, we find that the Bayesian evidences inferred with NPE ($\log p(x) = -7667.958\pm0.006$) and FMPE ($\log p(x) = -7667.969\pm0.005$) are consistent within their statistical uncertainties. A correct evidence is only obtained in importance sampling when the inferred posterior $q(\theta|x)$ covers the entire posterior $p(\theta|x)$ [41], so this is another indication that FMPE indeed induces mass-covering posteriors.

## 5.3 Discussion

Our results for GW150914 show that FMPE substantially outperforms NPE on this challenging problem. We believe that this is related to the network structure as follows. The NPE network allocates roughly two thirds of its parameters to the discrete normalizing flow and only one third to the embedding network (i.e., the feature extractor for $x$). Since FMPE parameterizes just a vector field (rather than a collection of splines in the normalizing flow) it can devote its network capacity to the interpretation of the high-dimensional $x \in \mathbb{R}^{15744}$. Hence, it scales better to larger networks and achieves higher accuracy. Remarkably, the performance iscomparable to GNPE, which involves a much simpler learning task with likelihood symmetries integrated by construction. This enhanced performance, comes in part at the cost of increased inference times, typically requiring hundreds of network forward passes. See Appendix D for further details.

In future work we plan to carry out a more complete analysis of GW inference using FMPE. Indeed, GW150914 is a loud event with good data quality, where NPE already performs quite well. DINGO with GNPE has been validated in a variety of settings [7, 69, 41, 74] including events with a larger performance gap between NPE and GNPE [69]. Since FMPE (like NPE) does not integrate physical symmetries, it would likely need further enhancements to fully compete with GNPE. This may require a symmetry-aware architecture [75], or simply further scaling to larger networks. A straightforward application of the GNPE mechanism to FMPE—GFMPE—is also possible, but less practical due to the higher inference costs of FMPE. Nevertheless, our results demonstrate that FMPE is a promising direction for future research in this field.

# 6 Conclusions

We introduced flow matching posterior estimation, a new simulation-based inference technique based on continuous normalizing flows. In contrast to existing neural posterior estimation methods, it does not rely on restricted density estimation architectures such as discrete normalizing flows, and instead parametrizes a distribution in terms of a conditional vector field. Besides enabling flexible path specifications, while maintaining direct access to the posterior density, we empirically found that regressing on a vector field rather than an entire distribution improves the scalability of FMPE compared to existing approaches. Indeed, fewer parameters are needed to learn this vector field, allowing for larger networks, ultimately enabling to solve more complex problems. Furthermore, our architecture for FMPE (a straightforward ResNet with GLU conditioning) facilitates parallelization and allows for cheap forward/backward passes.

We evaluated FMPE on a set of 10 benchmark tasks and found competitive or better performance compared to other simulation-based inference methods. On the challenging task of gravitational-wave inference, FMPE substantially outperformed comparable discrete flows, producing samples on par with a method that explicitly leverages symmetries to simplify training. Additionally, flow matching latent spaces are more naturally structured than those of discrete flows, particularly when using paths such as optimal transport. Looking forward, it would be interesting to exploit such structure in designing learning algorithms. This performance and flexibilty underscores the capability of continuous normalizing flows to efficiently solve inverse problems.

## Acknowledgements

We thank the DINGO team for helpful discussions and comments. We would like to particularly acknowledge the contributions of Alessandra Buonanno, Jonathan Gair, Nihar Gupte and Michael Pürrer. This material is based upon work supported by NSF's LIGO Laboratory which is a major facility fully funded by the National Science Foundation. This research has made use of data or software obtained from the Gravitational Wave Open Science Center (gw-openscience.org), a service of LIGO Laboratory, the LIGO Scientific Collaboration, the Virgo Collaboration, and KAGRA. LIGO Laboratory and Advanced LIGO are funded by the United States National Science Foundation (NSF) as well as the Science and Technology Facilities Council (STFC) of the United Kingdom, the Max-Planck-Society (MPS), and the State of Niedersachsen/Germany for support of the construction of Advanced LIGO and construction and operation of the GEO600 detector. Additional support for Advanced LIGO was provided by the Australian Research Council. Virgo is funded, through the European Gravitational Observatory (EGO), by the French Centre National de Recherche Scientifique (CNRS), the Italian Istituto Nazionale di Fisica Nucleare (INFN) and the Dutch Nikhef, with contributions by institutions from Belgium, Germany, Greece, Hungary, Ireland, Japan, Monaco, Poland, Portugal, Spain. The construction and operation of KAGRA are funded by Ministry of Education, Culture, Sports, Science and Technology (MEXT), and Japan Society for the Promotion of Science (JSPS), National Research Foundation (NRF) and Ministry of Science and ICT (MSIT) in Korea, Academia Sinica (AS) and the Ministry of Science and Technology (MoST) in Taiwan. M.D. thanks the Hector Fellow Academy for support. J.H.M. and B.S. are members of the MLCoE, EXC number 2064/1 – Project number 390727645 and the Tübingen AI Center funded by the German Ministry for Science and Education (FKZ 01IS18039A).

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

## A Gaussian flow

We here derive the form of a vector field $v_t(\theta)$ that restricts the resulting continuous flow to a one dimensional Gaussian with mean $\hat{\mu}$ variance $\hat{\sigma}^2$. With the optimal transport path $\mu_t(\theta) = t\theta_1$, $\sigma_t(\theta) = 1 - (1 - \sigma_{\min})t \equiv \sigma_t$ from [16], the sample-conditional probability path (5) reads

$$p_t(\theta|\theta_1) = \mathcal{N}[t\theta_1, \sigma_t^2](\theta). \tag{10}$$

We set our target distribution

$$q_1(\theta_1) = \mathcal{N}[\hat{\mu}, \hat{\sigma}^2](\theta_1). \tag{11}$$

To derive the marginal probability path and the marginal vector field we need two identities for the convolution $*$ of Gaussian densities. Recall that the convolution of two function is defined by $f * g(x) = \int f(x - y)g(y)\,\mathrm{d}y$. We define the function

$$g_{\mu,\sigma^2}(\theta) = \theta \cdot \mathcal{N}\left[\mu, \sigma^2\right](\theta). \tag{12}$$

Then the following holds

$$\mathcal{N}[\mu_1, \sigma_1^2] * \mathcal{N}[\mu_2, \sigma_2^2] = \mathcal{N}[\mu_1 + \mu_2, \sigma_1^2 + \sigma_2^2] \tag{13}$$

$$g_{0,\sigma_1^2} * \mathcal{N}[\mu_2, \sigma_2^2] = \frac{\sigma_1^2}{\sigma_1^2 + \sigma_2^2}\left(g_{\mu_2, \sigma_1^2 + \sigma_2^2} - \mu_2\,\mathcal{N}[\mu_2, \sigma_1^2 + \sigma_2^2]\right) \tag{14}$$

**Marginal probability paths**

Marginalizing over $\theta_1$ in (10) with (11), we find

$$
\begin{aligned}
p_t(\theta) &= \int p_t(\theta|\theta_1)q(\theta_1)\,d\theta_1 \\
&= \int \mathcal{N}\left[t\theta_1, \sigma_t^2\right](\theta)\,\mathcal{N}\left[\hat{\mu}, \hat{\sigma}^2\right](\theta_1)d\theta_1 \\
&= \int \mathcal{N}\left[0, \sigma_t^2\right](\theta - t\theta_1)\,\mathcal{N}(t\hat{\mu}, (t\hat{\sigma})^2)(t\theta_1) \cdot t\,d\theta_1 \\
&= \int \mathcal{N}\left[0, \sigma_t^2\right](\theta - \theta_1^t)\,\mathcal{N}\left[t\hat{\mu}, (t\hat{\sigma})^2\right](\theta_1^t)\,d\theta_1^t \\
&= \mathcal{N}\left[t\hat{\mu}, \sigma_t^2 + (t\hat{\sigma})^2\right](\theta)
\end{aligned} \tag{15}
$$

where we defined $\theta_1^t = t\theta_1$ and used (13).

**Marginal vector field**

We now calculate the marginalized vector field $u_t(\theta)$ based on equation (8) in [16]. Using the sample-conditional vector field (6) and the distributions (10) and (11) we find

$$
\begin{aligned}
u_t(\theta) &= \int u_t(\theta|\theta_1)\frac{p_t(\theta|\theta_1)q(\theta_1)}{p_t(\theta)}\,d\theta_1 \\
&= \frac{1}{p_t(\theta)}\int \frac{(\theta_1 - (1 - \sigma_{\min})\theta)}{\sigma_t} \cdot \mathcal{N}\left[t\theta_1, \sigma_t^2\right](\theta) \cdot \mathcal{N}\left[\hat{\mu}, \hat{\sigma}^2\right](\theta_1)\,d\theta_1 \\
&= \frac{1}{p_t(\theta)}\int \frac{(\theta_1 - (1 - \sigma_{\min})\theta)}{\sigma_t} \cdot \mathcal{N}\left[0, \sigma_t^2\right](\theta - t\theta_1) \cdot \mathcal{N}\left[t\hat{\mu}, (t\hat{\sigma})^2\right](t\theta_1) \cdot t\,d\theta_1 \\
&= \frac{1}{p_t(\theta)}\int \frac{(\theta_1' - (1 - \sigma_{\min})t \cdot \theta)}{\sigma_t \cdot t} \cdot \mathcal{N}\left[0, \sigma_t^2\right](\theta - \theta_1') \cdot \mathcal{N}\left[t\hat{\mu}, (t\hat{\sigma})^2\right](\theta_1') \cdot d\theta_1' \\
&= \frac{1}{p_t(\theta)}\int \frac{(-\theta_1'' + (1 - (1 - \sigma_{\min})t) \cdot \theta)}{\sigma_t \cdot t} \cdot \mathcal{N}\left[0, \sigma_t^2\right](\theta_1'') \cdot \mathcal{N}\left[t\hat{\mu}, (t\hat{\sigma})^2\right](\theta - \theta_1'') \cdot d\theta_1'' \\
&= \frac{1}{p_t(\theta)}\int \frac{(-\theta_1'' + \sigma_t \cdot \theta)}{\sigma_t \cdot t} \cdot \mathcal{N}\left[0, \sigma_t^2\right](\theta_1'') \cdot \mathcal{N}\left[t\hat{\mu}, (t\hat{\sigma})^2\right](\theta - \theta_1'') \cdot d\theta_1''
\end{aligned}
$$

$$\tag{16}$$

where we used the change of variables $\theta_1' = t\theta_1$ and $\theta_1'' = \theta - \theta_1'$. Now we evaluate this expression using (12), then the identities (13) and (14) and the marginal probability (15)

$$
\begin{aligned}
u_t(\theta) &= \frac{-1}{p_t(\theta) \cdot \sigma_t \cdot t} \left( g_{0,\sigma_t^2} * \mathcal{N}\left[t\hat{\mu}, (t\hat{\sigma})^2\right] \right)(\theta) + \frac{\theta}{p_t(\theta) \cdot t} \left( \mathcal{N}\left[0, \sigma_t^2\right] * \mathcal{N}\left[t\hat{\mu}, (t\hat{\sigma})^2\right] \right)(\theta) \\
&= \frac{-1}{p_t(\theta) \cdot \sigma_t \cdot t} \frac{(\theta - t\hat{\mu}) \cdot \sigma_t^2}{\sigma_t^2 + (t\hat{\sigma})^2} \cdot \mathcal{N}\left[t\hat{\mu}, (\sigma_t^2 + (t\hat{\sigma})^2)\right](\theta) + \frac{\theta}{p_t(\theta) \cdot t} \mathcal{N}\left[t\hat{\mu}, (\sigma_t^2 + (t\hat{\sigma})^2)\right](\theta) \\
&= \frac{(\sigma_t^2 + (t\hat{\sigma})^2)\theta - (\theta - t\hat{\mu}) \cdot \sigma_t}{p_t(\theta) \cdot t \cdot (\sigma_t^2 + (t\hat{\sigma})^2)} \cdot p_t(\theta) \\
&= \frac{(\sigma_t^2 + (t\hat{\sigma})^2 - \sigma_t)\theta + t\hat{\mu} \cdot \sigma_t}{t \cdot (\sigma_t^2 + (t\hat{\sigma})^2)}.
\end{aligned}
\tag{17}
$$

By choosing a vector field $v_t$ of the form (17) with learnable parameters $\hat{\mu}, \hat{\sigma}^2$, we can thus define a continuous flow that is restricted to a one dimensional Gaussian.

## B  Mass covering properties of flows

In this supplement, we investigate the mass covering properties of continuous normalizing flows trained using mean squared error and in particular prove Theorem 1. We first recall the notation from the main part. We always assume that the data is distributed according to $p_1(\theta)$. In addition, there is a known and simple base distribution $p_0$ and we assume that there is a vector field $u_t : [0,1] \times \mathbb{R}^d \to \mathbb{R}^d$ that connects $p_0$ and $p_1$ in the following sense. We denote by $\phi_t$ the flow generated by $u_t$, i.e., $\phi_t$ satisfies

$$
\partial_t \phi_t(\theta) = u_t(\phi_t(\theta)).
\tag{18}
$$

Then we assume that $(\phi_1)_* p_0 = p_1$ and we also define the interpolations $p_t = (\phi_t)_* p_0$.

We do not have access to the ground truth distributions $p_t$ and the vector field $u_t$ but we try to learn a vector field $v_t$ approximating $u_t$. We denote its flow by $\psi_t$ and we define $q_t = (\psi_t)_* q_0$ and $q_0 = p_0$. We are interested in the mass covering properties of the learned approximation $q_1$ of $p_1$. In particular, we want to relate the KL-divergence $\mathrm{D}_{\mathrm{KL}}(p_1 \| q_1)$ to the mean squared error,

$$
\mathrm{MSE}_p(u, v) = \int_0^1 \mathrm{d}t \int p_t(\mathrm{d}\theta)(u_t(\theta) - v_t(\theta))^2,
\tag{19}
$$

of the generating vector fields. The first observation is that without any regularity assumptions on $v_t$ it is impossible to obtain any bound on the KL-divergence in terms of the mean squared error.

**Lemma 1.** *For every $\varepsilon > 0$ there are vector field $u_t$ and $v_t$ and a base distribution $p_0 = q_0$ such that*

$$
\mathrm{MSE}_p(u, v) < \varepsilon \text{ and } \mathrm{D}_{\mathrm{KL}}(p_1 \| q_1) = \infty.
\tag{20}
$$

*In addition we can construct $u_t$ and $v_t$ such that the support of $p_1$ is larger than the support of $q_1$.*

*Proof.* We consider the uniform distribution $p_0 = q_0 \sim \mathcal{U}([-1, 1])$ and the vector fields

$$
u_t(\theta) = 0
\tag{21}
$$

and

$$
v_t(\theta) = \begin{cases} \varepsilon & \text{for } 0 \leq \theta < \varepsilon, \\ 0 & \text{otherwise.} \end{cases}
\tag{22}
$$

As before, let $\phi_t$ denote the flow of the vector field $u_t$ and similarly $\psi_t$ denote the flow of $v_t$. Clearly $\phi_t(\theta) = \theta$. On the other hand

$$
\psi_t(\theta) = \begin{cases} \min(\theta + \varepsilon t, \varepsilon) & \text{if } 0 \leq \theta < \varepsilon, \\ \theta & \text{otherwise.} \end{cases}
\tag{23}
$$

In particular

$$\psi_1(\theta) = \begin{cases} \varepsilon & \text{if } 0 \le \theta < \varepsilon, \\ \theta & \text{otherwise.} \end{cases} \tag{24}$$

This implies that $p_1 = (\phi_1)_* p_0 \sim \mathcal{U}([-1, 1])$. On the other hand $q_1 = (\psi_1)_* q_0$ has support in $[-1, 0] \cup [\varepsilon, 1]$. In particular, the distribution of $q_1$ is not mass covering with respect to $p_1$ and $D_{KL}(p_1 \| q_1) = \infty$. Finally, we observe that the MSE can be arbitrarily small

$$\text{MSE}_p(u, v) = \int_0^1 dt \int p_t(d\theta) |u_t(\theta) - v_t(\theta)|^2 = \int_0^1 \int_0^\varepsilon \frac{1}{2} \varepsilon^2 = \frac{\varepsilon^3}{2}. \tag{25}$$

Here we used that the density of $p_t(d\theta)$ is $1/2$ for $-1 \le \theta \le 1$. $\qquad\square$

We see that an arbitrary small MSE-loss cannot ensure that the probability distribution is mass covering and the KL-divergence is finite. On a high level this can be explained by the fact that for vector fields $v_t$ that are not Lipschitz continuous the flow is not necessarily continuous, and we can generate holes in the distribution. Note that we chose $p_0$ to be a uniform distribution for simplicity, but the result extends to any smooth distribution, in particular the result does not rely on the discontinuity of $p_0$.

Next, we investigate the mass covering property for Lipschitz continuous flows. When the flows $u_t$ and $v_t$ are Lipschitz continuous (in $\theta$) this ensures that the flows $\psi_1$ and $\phi_1$ are continuous in $x$ and it is not possible to create holes in the distribution as shown above for non-continuous vector fields. We show a weaker bound in this setting.

**Lemma 2.** *For every $0 \le \delta \le 1$ there is a base distribution $p_0 = q_0$ and the are Lipschitz-continuous vector fields $u_t$ and $v_t$ such that $\text{MSE}_p(u, v) = \delta$ and*

$$D_{KL}(p_1 \| q_1) \ge \frac{1}{2} \text{MSE}_p(u, v)^{1/3}. \tag{26}$$

*Proof.* We consider $p_0$, $q_0$ and $u_t$ as in Lemma 1, and we define

$$v_t(\theta) = \begin{cases} 2\theta & \text{for } 0 \le \theta < \varepsilon, \\ 2\varepsilon - \theta & \text{for } \varepsilon \le \theta < 2\varepsilon, \\ 0 & \text{otherwise.} \end{cases} \tag{27}$$

Then we can calculate for $0 \le \theta \le e^{-2}\varepsilon$ that

$$\psi_t(\theta) = \theta e^{2t}. \tag{28}$$

Similarly we obtain for $\varepsilon \le \theta \le 2\varepsilon$ (solving the ODE $f' = 2f$)

$$\psi_t(\theta) = 2\varepsilon - (2\varepsilon - \theta)e^{-2t}. \tag{29}$$

We find

$$\psi_1(0) = 0, \ \psi_1(e^{-2}\varepsilon) = \varepsilon, \ \psi_1(\varepsilon) = 2 - \varepsilon e^{-2}; \ \psi_2(2\varepsilon) = 2\varepsilon. \tag{30}$$

Next we find for the densities of $q_1$ that

$$q_1(\psi_1(\theta)) = q_0(\theta) |\psi_1'(\theta)|^{-1} = \frac{1}{2} \begin{cases} e^{-2} & \text{for } 0 \le \theta \le e^{-2}\varepsilon, \\ e^2 & \text{for } \varepsilon \le \theta \le 2\varepsilon. \end{cases} \tag{31}$$

Together with (30) this implies that the density of $q_1$ is given by

$$q_1(\theta) = \frac{1}{2} \begin{cases} e^{-2} & \text{for } 0 \le \theta \le \varepsilon, \\ e^2 & \text{for } 2\varepsilon - \varepsilon e^{-2} \le \theta \le 2\varepsilon. \end{cases} \tag{32}$$

Note that $p_1(\theta) = 1/2$ for $-1 \le \theta \le 1$ and therefore

$$\int_0^\varepsilon \ln \frac{p_1(\theta)}{q_1(\theta)} p_1(d\theta) = \int_0^\varepsilon \ln(e^2) \frac{1}{2} d\theta = \varepsilon, \tag{33}$$

and

$$\int_{2\varepsilon-\varepsilon e^{-2}}^{2\varepsilon} \ln \frac{p_1(\theta)}{q_1(\theta)} p_1(\mathrm{d}\theta) = \int_{2\varepsilon-\varepsilon e^{-2}}^{2\varepsilon} \ln(e^{-2})\frac{1}{2}\mathrm{d}\theta = -\varepsilon e^{-2}. \tag{34}$$

Moreover we note

$$\int_{\varepsilon}^{2\varepsilon-\varepsilon e^{-2}} q_1(\mathrm{d}\varepsilon) = \int_{e^{-2}\varepsilon}^{\varepsilon} q_0(\mathrm{d}\varepsilon) = \frac{1}{2}\varepsilon(1-e^{-2}) = \int_{\varepsilon}^{2\varepsilon-\varepsilon e^{-2}} p_1(\mathrm{d}\varepsilon), \tag{35}$$

which implies (by positivity of the KL-divergence) that

$$\int_{\varepsilon}^{2\varepsilon-\varepsilon e^{-2}} \ln\left(\frac{p_1(\theta)}{q_1(\theta)}\right) p_1(\mathrm{d}\theta) \geq 0. \tag{36}$$

We infer using also $p_1(\theta) = q_1(\theta) = 1/2$ for $\theta \in [-1,0] \cap [2\varepsilon,1]$ that

$$\mathrm{D_{KL}}(p_1||q_1) = \int \ln\left(\frac{p_1(\theta)}{q_1(\theta)}\right) p_1(\mathrm{d}\theta) \geq \varepsilon(1-e^{-2}). \tag{37}$$

On the other hand we can bound

$$\int_0^1 \mathrm{d}t \int p_t(\mathrm{d}\theta)|v_t(\theta) - u_t(\theta)|^2 = \frac{1}{2}\int_0^1 \mathrm{d}t \int_0^{2\varepsilon} |u_t(\theta)|^2 = \int_0^{\varepsilon} s^2 \, \mathrm{d}s = \frac{\varepsilon^3}{3}. \tag{38}$$

We conclude that

$$\mathrm{D_{KL}}(p_1||q_1) \geq \frac{1}{2}\left(\mathrm{MSE}_p(u,v)\right)^{1/3}. \tag{39}$$

In particular, it is not possible to bound the KL-divergence by the MSE even when the vector fields are Lipschitz continuous. $\qquad\square$

Let us put this into context. It was already shown in [42] that we can, in general, not bound the forward KL-divergence by the mean squared error and our Lemmas 1 and 2 are concrete examples. On the other hand, when considering SDEs the KL-divergence can be bounded by the mean squared error of the drift terms as shown in [43]. Indeed, in [42] the favorable smoothing effect was carefully investigated.

Here we show that we can alternatively obtain an upper bound on the KL-divergence when assuming that $u_t$, $v_t$, and $p_0$ satisfy additional regularity assumptions. This allows us to recover the mass covering property from bounds on the means squared error for sufficiently smooth vector fields. The scaling is nevertheless still weaker than for SDEs.

We now state our assumptions. We denote the gradient with respect to $\theta$ by $\nabla = \nabla_\mu$ and second derivatives by $\nabla^2 = \nabla^2_{\mu\nu}$. When applying the chain rule, we leave the indices implicit. We denote by $|\cdot|$ the Frobenius norm $|A| = \left(\sum_{ij} A_{ij}^2\right)^{1/2}$ of a matrix. The Frobenius norm is submultiplicative, i.e., $|AB| \leq |A| \cdot |B|$ and directly generalizes to higher order tensors.

*Assumption* 1. We assume that

$$|\nabla u_t| \leq L, \; |\nabla v_t| \leq L, \; |\nabla^2 u_t| \leq L', \; |\nabla^2 v_t| \leq L'. \tag{40}$$

We require one further assumption on $p_0$.

*Assumption* 2. There is a constant $C_1$ such that

$$|\nabla \ln p_0(\theta)| \leq C_1(1 + |\theta|). \tag{41}$$

We also assume that

$$\mathbb{E}_{p_0} |\theta|^2 < C_2 < \infty. \tag{42}$$

Note that (41) holds, e.g., if $p_0$ follows a Gaussian distribution but also for smooth distribution with slower decay at $\infty$. If we assume that $|\nabla \ln p_0(\theta)|$ is bounded the proof below simplifies slightly. This is, e.g., the case if $p_0(\theta) \sim e^{-|\theta|}$ as $|\theta| \to \infty$.

We need some additional notation. It is convenient to introduce $\phi_t^s = \phi_t \circ (\phi_s)^{-1}$, i.e., the flow from time $s$ to $t$ (in particular $\phi_t^0 = \phi_t$) and similarly for $\psi$. We can now restate and prove Theorem 1.

**Theorem 2.** *Let $p_0 = q_0$ and assume $u_t$ and $v_t$ are two vector fields whose flows satisfy $p_1 = (\phi_1)_* p_0$ and $q_1 = (\psi_1)_* q_0$. Assume that $p_0$ satisfies Assumption 2 and $u_t$ and $v_t$ satisfy Assumption 1. Then there is a constant $C > 0$ depending on $L$, $L'$, $C_1$, $C_2$, and $d$ such that (for $\mathrm{MSE}_p(u,v) < 1$)*

$$D_{\mathrm{KL}}(p_1 \| q_1) \leq C \, \mathrm{MSE}_p(u,v)^{\frac{1}{2}}. \tag{43}$$

*Remark* 1. We do not claim that our results are optimal, it might be possible to find similar bounds for the forward KL-divergence with weaker assumptions. However, we emphasize that Lemma 2 shows that the result of the theorem is not true without the assumption on the second derivative of $v_t$ and $u_t$.

*Proof.* We want to control $D_{\mathrm{KL}}(p_1 \| q_1)$. It can be shown that (see equation above (25) in [43] or Lemma 2.19 in [42] )

$$\partial_t D_{\mathrm{KL}}(p_t \| q_t) = -\int p_t(\mathrm{d}\theta)(u_t(\theta) - v_t(\theta)) \cdot (\nabla \ln p_t(\theta) - \nabla \ln q_t(\theta)). \tag{44}$$

Using Cauchy-Schwarz we can bound this by

$$\partial_t D_{\mathrm{KL}}(p_t \| q_t) \leq \left( \int p_t(\mathrm{d}\theta) |u_t(\theta) - v_t(\theta)|^2 \right)^{\frac{1}{2}} \left( \int p_t(\mathrm{d}\theta) |\nabla \ln p_t(\theta) - \nabla \ln q_t(\theta)|^2 \right)^{\frac{1}{2}}. \tag{45}$$

We use the relation (see (3))

$$\ln(p_t(\phi_t(\theta_0)) = \ln(p_0(\theta_0)) - \int_0^t (\mathrm{div}\, u_s)(\phi_s(\theta_0))\mathrm{d}s, \tag{46}$$

which can be equivalently rewritten (setting $\theta = \phi_t \theta_0$) as

$$\ln(p_t(\theta)) = \ln(p_0(\phi_0^t \theta)) - \int_0^t (\mathrm{div}\, u_s)(\phi_s^t \theta)\mathrm{d}s. \tag{47}$$

We use the following relation for $\nabla \phi_s^t$

$$\nabla \phi_s^t(\theta) = \exp\left( \int_t^s \mathrm{d}\tau \, (\nabla u_\tau)(\phi_\tau^t(\theta)) \right). \tag{48}$$

This relation is standard and can be directly deduced from the following ODE for $\nabla \phi_s^t$

$$\partial_s \nabla \phi_s^t(\theta) = \nabla \partial_s \phi_s^t(\theta) = \nabla(u_s(\phi_s^t(\theta))) = \left( (\nabla u_s)(\phi_s^t(\theta)) \right) \cdot \nabla \phi_s^t(\theta). \tag{49}$$

We can conclude that for $0 \leq s, t \leq 1$ the bound

$$|\nabla \phi_s^t(\theta)| \leq e^L \tag{50}$$

holds. We find

$$
\begin{aligned}
|\nabla \ln(p_t(\theta))| &= \left| \nabla \ln(p_0)(\phi_0^t \theta) \cdot \nabla \phi_0^t(\theta) - \int_0^t (\nabla \, \mathrm{div}\, u_s)(\phi_s^t \theta) \cdot \nabla \phi_s^t(\theta)\mathrm{d}s \right| \\
&\leq |\nabla \ln(p_0)(\phi_0^t \theta)| e^L + L' e^L,
\end{aligned} \tag{51}
$$

and a similar bound holds for $q_t$. In words, we have shown that the score of $p_t$ at $\theta$ can be bounded by the score of $p_0$ of theta transported along the vector field $u_t$ minus a correction which quantifies the change of score along the path. We now bound using the definition $p_t = (\phi_t)_* p_0$ and the assumption (41)

$$
\begin{aligned}
\int p_t(\mathrm{d}\theta) |\nabla \ln p_0(\phi_0^t(\theta))|^2 &= \int p_0(\mathrm{d}\theta_0) |\nabla \ln p_0(\phi_0^t \phi_t(\theta_0))|^2 = \mathbb{E}_{p_0} |\nabla \ln p_0(\theta_0)|^2 \\
&\leq \mathbb{E}_{p_0}(C_1(1 + |\theta_0|)^2) \leq 2C_1^2(1 + \mathbb{E}_{p_0} |\theta_0|^2) \leq 2C_1^2(1 + C_2^2).
\end{aligned} \tag{52}
$$

Similarly we obtain using $q_0 = p_0$

$$\int p_t(\mathrm{d}\theta) |\nabla \ln q_0(\psi_0^t \theta)|^2 = \int p_0(\mathrm{d}\theta_0) |\nabla \ln q_0(\psi_0^t \phi_t \theta_0)|^2. \tag{53}$$

In words, to control the score of $q$ integrated with respect to $p_t$ we need to control the distortion we obtain when moving forward with $u$ and backwards with $v$. We investigate $\psi_0^t \phi_t(\theta_0)$. We now show

$$\partial_h \psi_t^{t+h} \phi_{t+h}^t(\theta)|_{h=0} = u_t(\theta) - v_t(\theta). \tag{54}$$

First, by definition of $\phi$, we find

$$\partial_h \phi_{t+h}^t(\theta)|_{h=0} = \partial_h \phi_{t+h} \phi_t^{-1}(\theta)|_{h=0} = u_t(\phi_t \phi_t^{-1}(\theta)) = u_t(\theta). \tag{55}$$

To evaluate the second contribution we observe

$$0 = \partial_h \theta|_{h=0} = \partial_h \psi_{t+h}^{t+h}(\theta)|_{h=0} = \partial_h \psi_{t+h} \psi_{t+h}^{-1}(\theta)|_{h=0}$$
$$= (\partial_h \psi_{t+h})\psi_t^{-1}(\theta)|_{h=0} + \psi_t(\partial_h \psi_{t+h}^{-1})(\theta)|_{h=0} = v_t(\psi_t \psi_t^{-1}(\theta)) + \partial_h \psi_t \psi_{t+h}^{-1}(\theta)|_{h=0} \tag{56}$$
$$= v_t(\theta) + \partial_h \psi_t^{t+h}(\theta)|_{h=0}$$

Now (54) follows from (55) and (56) together with $\phi_t^t = \psi_t^t = \mathrm{Id}$. Using (54) we find

$$\partial_t(\psi_0^t \phi_t)(\theta_0) = \partial_h(\psi_0^t \psi_t^{t+h} \phi_{t+h}^t \phi_t)(\theta_0)|_{h=0} = (\nabla \psi_0^t)(\phi_t(\theta_0)) \cdot ((u_t - v_t)(\phi_t(\theta_0))). \tag{57}$$

Using (50) we conclude that

$$|\psi_0^t \phi_t(\theta_0) - \theta_0| \le \left| \int_0^t \partial_s \psi_0^s \phi_s(\theta_0) \, ds \right| \le \int_0^t |(\nabla \psi_0^s)(\phi_s(\theta_0))| \cdot |u_s - v_s|(\phi_s(\theta_0)) \, ds$$
$$\le e^L \int_0^t |u_s - v_s|(\phi_s(\theta_0)) \, ds. \tag{58}$$

We use this and the assumption (41) to continue to estimate (53) as follows

$$\int p_t(\mathrm{d}\theta)|\nabla \ln q_0(\psi_0^t \theta)|^2 = \int p_0(\mathrm{d}\theta_0)|\nabla \ln q_0(\psi_0^t \phi_t(\theta_0))|^2$$
$$\le C_1^2 \int p_0(\mathrm{d}\theta_0)(1 + |\psi_0^t \phi_t(\theta_0)|)^2$$
$$\le C_1^2 \int p_0(\mathrm{d}\theta_0)(1 + |\psi_0^t \phi_t(\theta_0) - \theta_0| + |\theta_0|)^2$$
$$\le 3C_1^2 + 3C_1^2 \int p_0(\mathrm{d}\theta_0) \left( |\psi_0^t \phi_t(\theta_0) - \theta_0|^2 + |\theta_0|^2 \right)$$
$$\le 3C_1^2(1 + \mathbb{E}_{p_0} |\theta_0|^2) + 3C_1^2 e^{2L} \int p_0(\mathrm{d}\theta_0) \left( \int_0^t \mathrm{d}s \, |u_s - v_s|(\phi_s(\theta_0)) \right)^2. \tag{59}$$

Here we used $(a + b + c)^2 \le 3(a^2 + b^2 + c^2)$ in the second to last step. We bound the remaining integral using Cauchy-Schwarz as follows

$$\int p_0(\mathrm{d}\theta_0) \left( \int_0^t |u_s - v_s|(\phi_s(\theta_0)) \right)^2 \le \int p_0(\mathrm{d}\theta_0) \left( \int_0^t \mathrm{d}s \, |u_s - v_s|^2(\phi_s(\theta_0)) \right) \left( \int_0^t \mathrm{d}s \, 1^2 \right)$$
$$\le t \int_0^t \mathrm{d}s \int p_0(\mathrm{d}\theta_0)|u_s - v_s|^2(\phi_s(\theta_0))$$
$$= t \int_0^t \mathrm{d}s \int p_s(\mathrm{d}\theta_s)|u_s - v_s|^2(\theta_s)$$
$$\le \int_0^1 \mathrm{d}s \int p_s(\mathrm{d}\theta_s)|u_s - v_s|^2(\theta_s) = \mathrm{MSE}_p(u, v). \tag{60}$$

The last displays together imply

$$\int p_t(\mathrm{d}\theta)|\nabla \ln q_0(\psi_0^t \theta)|^2 \le 3C_1^2 \left( 1 + \mathbb{E}_{p_0} |\theta_0|^2 + e^{2L} \mathrm{MSE}_p(u, v) \right). \tag{61}$$

Now we have all the necessary ingredients to bound the derivative of the KL-divergence. We control the second integral in (45) using (51) (and again $(\sum_{i=1}^{4} a_i)^2 \leq 4 \sum a_i^2$) as follows,

$$\int p_t(\mathrm{d}\theta)|\nabla \ln p_t(\theta) - \nabla \ln q_t(\theta)|^2$$
$$\leq 2 \cdot 2^2 \cdot L'^2 e^{2L} + 4e^{2L} \int p_t(\mathrm{d}\theta) \left(|\nabla \ln q_0(\psi_0^t)\theta)|^2 + |\nabla \ln p_0(\phi_0^t)\theta)|^2\right). \tag{62}$$

Using (52) and (61) we finally obtain

$$\int p_t(\mathrm{d}\theta)|\nabla \ln p_t(\theta) - \nabla \ln q_t(\theta)|^2 \leq 8 \cdot L'^2 e^{2L} + C_1^2 e^{2L} \left(20(1 + C_2^2) + 12\,\mathrm{MSE}_p(u, v)\right)$$
$$\leq C(1 + \mathrm{MSE}_p(u, v)) \tag{63}$$

for some constant $C > 0$. Finally, we obtain

$$\mathrm{D}_{\mathrm{KL}}(p_1 || q_1) = \int_0^1 \mathrm{d}t \, \partial_t \mathrm{D}_{\mathrm{KL}}(p_t || q_t)$$
$$\leq (C(1 + \mathrm{MSE}_p(u, v)))^{\frac{1}{2}} \int_0^1 \mathrm{d}t \left(\int p_t(\mathrm{d}\theta)|u_t(\theta) - v_t(\theta)|^2\right)^{\frac{1}{2}}$$
$$\leq (C(1 + \mathrm{MSE}_p(u, v)))^{\frac{1}{2}} \left(\int_0^1 \mathrm{d}t \int p_t(\mathrm{d}\theta)|u_t(\theta) - v_t(\theta)|^2\right)^{\frac{1}{2}} \tag{64}$$
$$\leq (C(1 + \mathrm{MSE}_p(u, v)))^{\frac{1}{2}} \mathrm{MSE}_p(u, v)^{\frac{1}{2}}.$$

$\square$

## C  SBI Benchmark

In this section, we collect missing details and additional results for the analysis of the SBI benchmark in Section 4.

### C.1  Network architecture and hyperparameters

For each task and simulation budget in the benchmark, we perform a mild hyperparameter optimization. We sweep over the batch size and learning rate (which is particularly important as the simulation budgets differ by orders of magnitudes), the network size and the $\alpha$ parameter for the time prior defined in Section 3.3 (see Tab. 2 for the specific values). We reserve 5% of the simulation budget for validation and choose the model with the best validation loss across all configurations.

### C.2  Additional results

We here provide various additional results for the SBI benchmark. First, we compare the performance of FMPE and NPE when using the Maximum Mean Discrepancy metric (MMD). The results can be found in Fig. 6. FMPE shows superior performance to NPE for most tasks and simulation budgets. Compared to the C2ST scores in Fig. 4 the improvement shown by FMPE in MMD is more substantial.

Fig. 7 compares the FMPE results with the optimal transport path from the main text with a comparable score matching model using the Variance Preserving diffusion path [15]. The score matching results were obtained using the same batch size, network size and learning rate as the FMPE network, while optimizing for $\beta_{\min} \in \{0.1, 1, 4\}$ and $\beta_{\max} \in \{4, 7, 10\}$. FMPE with the optimal transport path clearly outperforms the score-based model on almost all configurations.

In Fig. 8 we compare FMPE using the architecture proposed in Section 3.2 with $(t, \theta)$-conditioning via gated linear units to FMPE with a naive architecture operating directly on the concatenated $(t, \theta, x)$ vector. For the two displayed tasks the context dimension $\dim(x) = 100$ is much larger than the parameter dimension $\dim(\theta) \in \{5, 10\}$, and there is a clear performance gain in using the GLU conditioning. Our interpretation is that the low dimensionality of $(t, \theta)$ means that it is not well-learned by the network when simply concatenated with $x$.

| hyperparameter | sweep values |
| --- | --- |
| hidden dimensions | $2^n$ for $n \in \{4, \ldots, 10\}$ |
| number of blocks | $10, \ldots, 18$ |
| batch size | $2^n$ for $n \in \{2, \ldots, 9\}$ |
| learning rate | 1.e-3, 5.e-4, 2.e-4, 1.e-4 |
| $\alpha$ (for time prior) | -0.25, -0.5, 0, 1, 4 |

Table 2: Sweep values for the hyperparamters for the SBI benchmark. We split the configurations according to simulation budgets, e.g. for 1000 simulations, we only swept over smaller values for network size and batch size. The network architecture has a diamond shape, with increasing layer width from smallest to largest and then decreasing to the output dimension. Each block consists of two fully-connected residual layers.

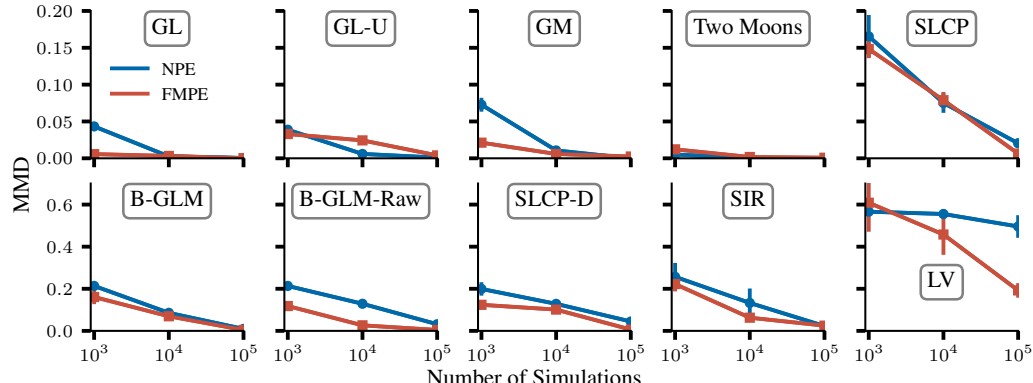

Figure 6: Comparison of FMPE and NPE performance across 10 SBI benchmarking tasks [46]. We here quantify the deviation in terms of the Maximum Mean Discrepancy (MMD) as an alternative metric to the C2ST score used in Fig. 4. MMD can be sensitive to its hyperparameters [46], so we use the C2ST score as a primary performance metric.

Fig. 9 displays the densities of the reference samples under the FMPE model as a histogram for all tasks (extended version of Fig. 3). The support of the learned model $q(\theta|x)$ covers the reference samples $\theta \sim p(\theta|x)$, providing additional empirical evidence for the mass-covering behavior theoretically explored in Thm. 1. However, samples from the true posterior distribution may have a small density under the learned model, especially if the deviation between model and reference is high; see Lotka-Volterra (bottom right panel). Fig. 10 displays P–P plots for two selected tasks.

Finally, we study the impact of our time prior re-weighting for one example task in Fig. 11. We clearly see that our proposed re-weighting leads to increased performance by up-weighting samples for $t$ closer to 1 during training.

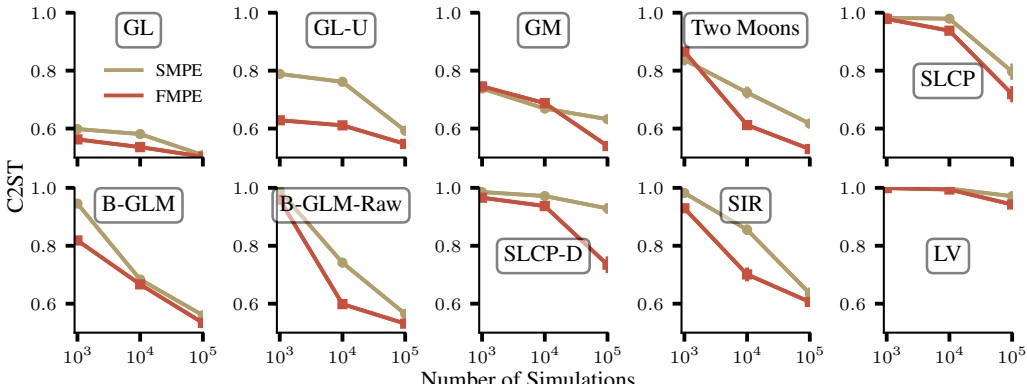

Figure 7: Comparison of FMPE with the optimal transport path (as used throughout the main paper) with comparable models trained with a Variance Preserving diffusion path [15] by regressing on the score (SMPE). Note that the SMPE baseline shown here is not directly comparable to NPSE [8, 9], as this method uses Langevin steps, which reduces the dependence of the results on the vector field for small $t$ (at the cost of a tractable density).

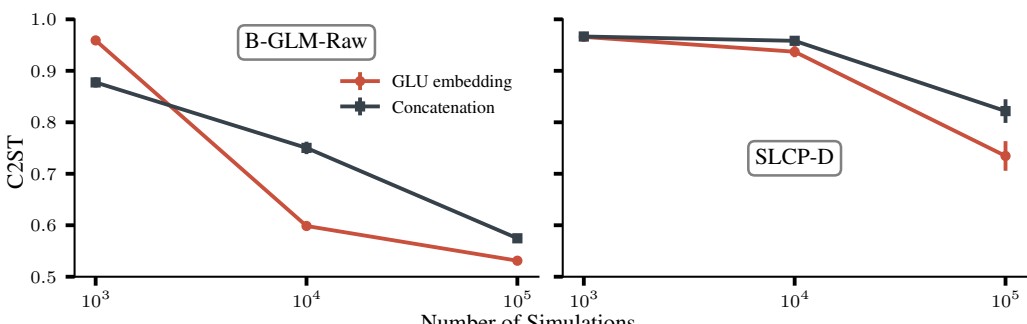

Figure 8: Comparison of the architecture proposed in Section 3.2 with gated linear units for the $(t, \theta)$-conditioning (red) and a naive architecture based on a simple concatenation of $(t, \theta, x)$ (black). FMPE with the proposed architecture performs substantially better.

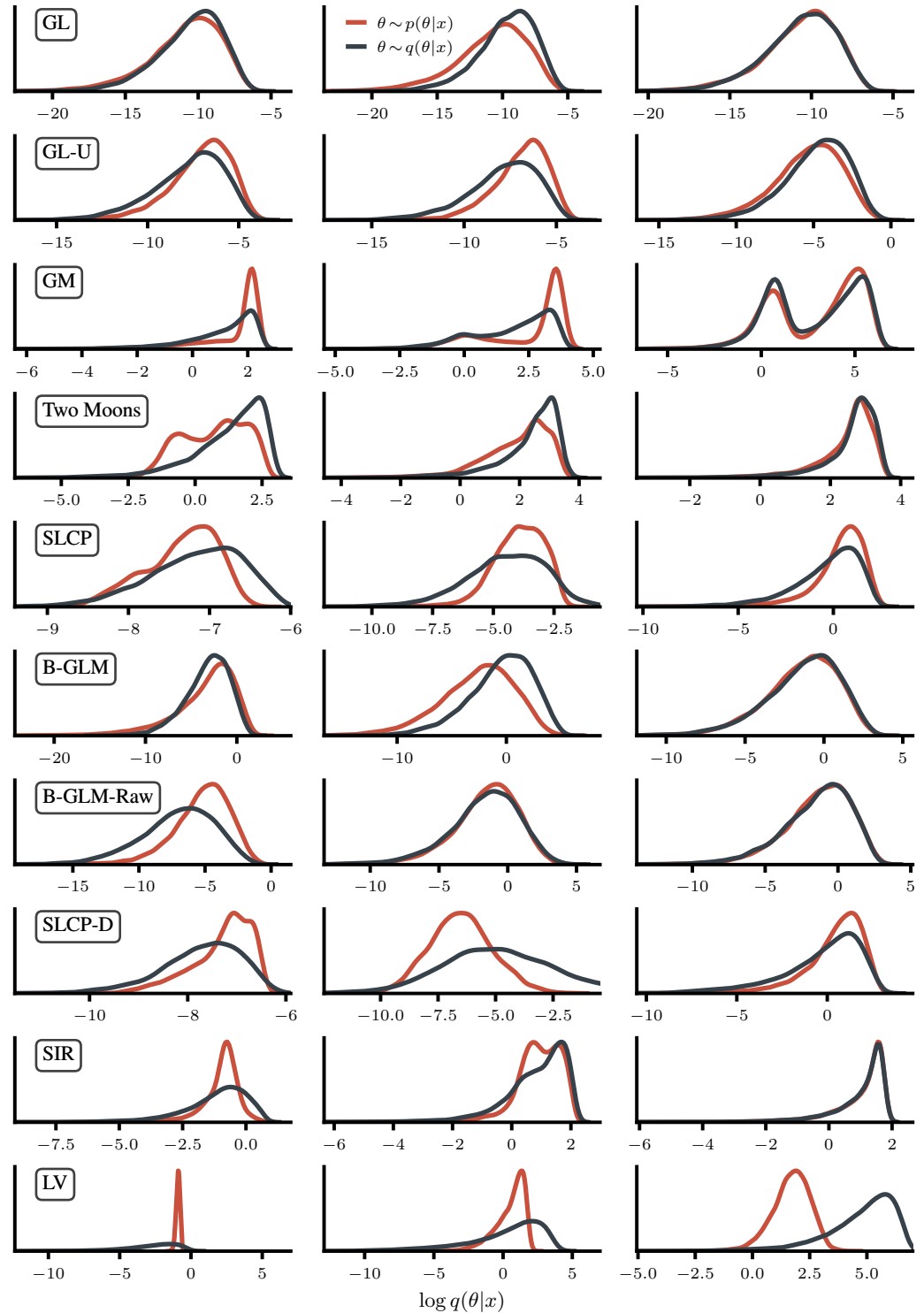

Figure 9: Histogram of FMPE densities $\log q(\theta|x)$ for samples $\theta \sim q(\theta|x)$ and reference samples $\theta \sim p(\theta|x)$ for simulation budgets $N = 10^3$ (left), $N = 10^4$ (center) and $N = 10^5$ (right). The reference samples $\theta \sim p(\theta|x)$ are all within the support of the learned model $q(\theta|x)$, indicating mass covering FMPE results. Nonetheless, reference samples may have a small density under $q(\theta|x)$, if the validation loss is high, see Lotka-Volterra (LV).

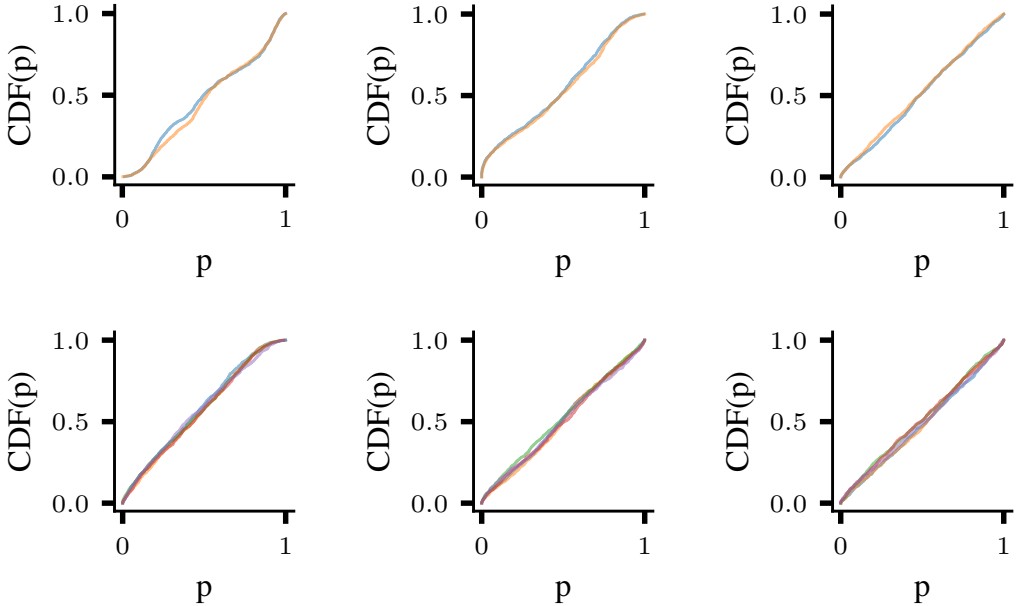

Figure 10: P-P plot for the marginals of the FMPE-posterior for the Two Moons (upper) and SLCP (lower) tasks for training budgets of $10^3$ (left), $10^4$ (center), and $10^5$ (right) samples.

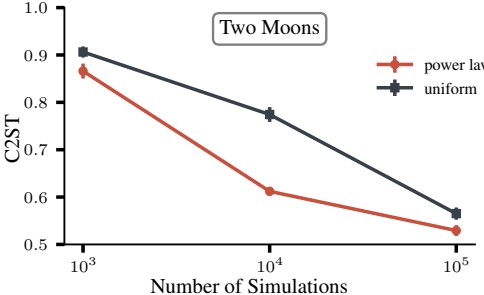

Figure 11: Comparison of the time prior re-weighting proposed in Section 3.3 with a uniform prior over $t$ on the Two Moons task (Section 4). The network trained with the re-weighted prior clearly outperforms the reference on all simulation budgets.

# D  Gravitational-wave inference

We here provide the missing details and additional results for the gravitational wave inference problem analyzed in Section 5.

## D.1  Network architecture and hyperparameters

Compared to NPE with normalizing flows, FMPE allows for generally simpler architectures, since the output of the network is simply a vector field. This also holds for NPSE (model also defined by a vector) and NRE (defined by a scalar). Our FMPE architecture builds on the embedding network developed in [7], however we extend the network capacity by adding more residual blocks (Tab. 3, top panel). For the $(t, \theta)$-conditioning we use gated linear units applied to each residual block, as described in Section 3.2. We also use a small residual network to embed $(t, \theta)$ before applying the gated linear units.

In this Appendix we also perform an ablation study, using the *same* embedding network as the NPE network (Tab. 3, bottom panel). For this configuration, we additionally study the effect of conditioning on $(t, \theta)$ starting from different layers of the main residual network.

## D.2  Data settings

We use the data settings described in [7], with a few minor modifications. In particular, we use the waveform model IMRPhenomPv2 [76–78] and the prior displayed in Tab. 4. Generation of the training dataset with 5,000,000 samples takes around 1 hour on 64 CPUs. Compared to [7], we reduce the frequency range from $[20, 1024]$ Hz to $[20, 512]$ Hz to reduce the computational load for data preprocessing. We also omit the conditioning on the detector noise power spectral density (PSD) introduced in [7] as we evaluate on a single GW event. Preliminary tests show that the performance with PSD conditioning is similar to the results reported in this paper. All changes to the data settings have been applied to FMPE and the NPE baselines alike to enable a fair comparison.

## D.3  Additional results

Tab. 5 displays the inference times for FMPE and NPE. NPE requires only a single network pass to produce samples and (log-)probabilities, whereas many forwards passes are needed for FMPE to solve the ODE with a specific level of accuracy. A significant portion of the additional time required for calculating (log-)probabilities in conjunction with the samples is spent on computing the divergence of the vector field, see Eq. (3).

| hyperparameter | values |
|---|---|
| residual blocks | $2048, 4096 \times 3, 2048 \times 3, 1024 \times 6, 512 \times 8, 256 \times 10,$ $128 \times 5, 64 \times 3, 32 \times 3, 16 \times 3$ |
| residual blocks $(t, \theta)$ embedding | $16, 32, 64, 128, 256$ |
| batch size | 4096 |
| learning rate | 5.e-4 |
| $\alpha$ (for time prior) | 1 |
| residual blocks | $2048 \times 2, 1024 \times 4, 512 \times 4, 256 \times 4, 128 \times 4, 64 \times 3,$ $32 \times 3, 16 \times 3$ |
| residual blocks $(t, \theta)$ embedding | $16, 32, 64, 128, 256$ |
| batch size | 4096 |
| learning rate | 5.e-4 |
| $\alpha$ (for time prior) | 1 |

Table 3: Hyperparameters for the FMPE models used in the main text (top) and in the ablation study (bottom, see Fig. 12). The network is composed of a sequence of residual blocks, each consisting of two fully-connected hidden layers, with a linear layer between each pair of blocks. The ablation network is the same as the embedding network that feeds into the NPE normalizing flow.

| Description | Parameter | Prior |
|---|---|---|
| component masses | $m_1, m_2$ | $[10, 120]$ M$_\odot$, $m_1 \geq m_2$ |
| chirp mass | $M_c = (m_1 m_2)^{\frac{3}{5}}/(m_1 + m_2)^{\frac{1}{5}}$ | $[20, 120]$ M$_\odot$ (constraint) |
| mass ratio | $q = m_2/m_1$ | $[0.125, 1.0]$ (constraint) |
| spin magnitudes | $a_1, a_2$ | $[0, 0.99]$ |
| spin angles | $\theta_1, \theta_2, \phi_{12}, \phi_{JL}$ | standard as in [79] |
| time of coalescence | $t_c$ | $[-0.03, 0.03]$ s |
| luminosity distance | $d_L$ | $[100, 1000]$ Mpc |
| reference phase | $\phi_c$ | $[0, 2\pi]$ |
| inclination | $\theta_{JN}$ | $[0, \pi]$ uniform in sine |
| polarization | $\psi$ | $[0, \pi]$ |
| sky position | $\alpha, \beta$ | uniform over sky |

Table 4: Priors for the astrophysical binary black hole parameters. Priors are uniform over the specified range unless indicated otherwise. Our models infer the mass parameters in the basis $(M_c, q)$ and marginalize over the phase parameter $\phi_c$.

Fig. 12 presents a comparison of the FMPE performance using networks of the same hidden dimensions as the NPE embedding network (Tab. 3 bottom panel). This comparison includes an ablation study on the timing of the $(t, \theta)$ GLU-conditioning. In the top-row network, the $(t, \theta)$ conditioning is applied only after the 256-dimensional blocks. In contrast, the middle-row network receives $(t, \theta)$ immediately after the initial residual block. With FMPE we can achieve performance comparable to NPE, while having only $\approx 1/3$ of the network size (most of the NPE network parameters are in the flow). This suggests that parameterizing the target distribution in terms of a vector field requires less learning capacity, compared to directly learning its density. Delaying the $(t, \theta)$ conditioning until the final layers impairs performance. However, the number of FLOPs at inference is considerably reduced, as the context embedding can be cached and a network pass only involves the few layers with the $(t, \theta)$ conditioning. Consequently, there's a trade-off between accuracy and inference speed, which we will explore in a greater scope in future work.

|  | Network Passes | Inference Time (per batch) |
|---|---|---|
| FMPE (sample only) | 248 | 26s |
| FMPE (sample and log probs) | 350 | 352s |
| NPE (sample and log probs) | 1 | 1.5s |

Table 5: Inference times per batch for FMPE and NPE on a single Nvidia A100 GPU, using the training batch size of 4096. We solve the ODE for FMPE using the `dopri5` discretization [80] with absolute and relative tolerances of 1e-7. For FMPE, generation of the (log-)probabilities additionally requires the computation of the divergence, see equation (3). This needs additional memory and therefore limits the maximum batch size that can be used at inference.

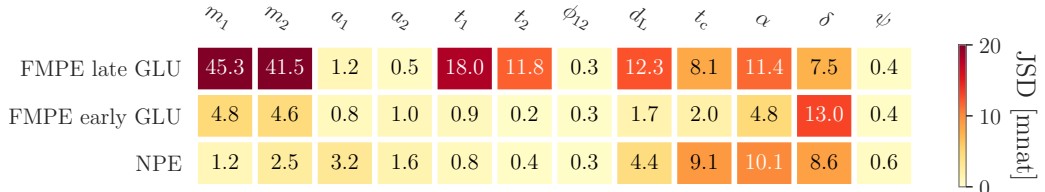

Figure 12: Jensen-Shannon divergence between inferred posteriors and the reference posteriors for GW150914 [73]. We compare two FMPE models with the same architecture as the NPE embedding network, see Tab. 3 bottom panel. For the model in the first row, the GLU conditioning of $(\theta, t)$ is only applied before the final 128-dim blocks. The model in the middle row is given the context after the very first 2048 block.

