# Flow Matching for Scalable Simulation-Based Inference

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

}$, and thereby scales much better to larger networks and achieve much higher accuracy. Remarkably, the performance is even comparable to GNPE, which involves a much simpler learning task with likelihood symmetries integrated by construction. See Appendix D for further details.

---

[5]Our implementation builds on the public DINGO code from `https://github.com/dingo-gw/dingo`.

[6]We omit the three parameters $\phi_c, \phi_{JL}, \theta_{JN}$ in the evaluation as we use phase marginalization in importance sampling and the reference therefore uses a different basis for these parameters [40]. For GNPE we report the results from [7], which are generated with slightly different data conditioning. Therefore, we do not display the GNPE results in the corner plot, and the JSDs serve only as a rough comparison. The JSD for the $t_c$ parameter is not reported in [7] due to a $t_c$ marginalized reference.

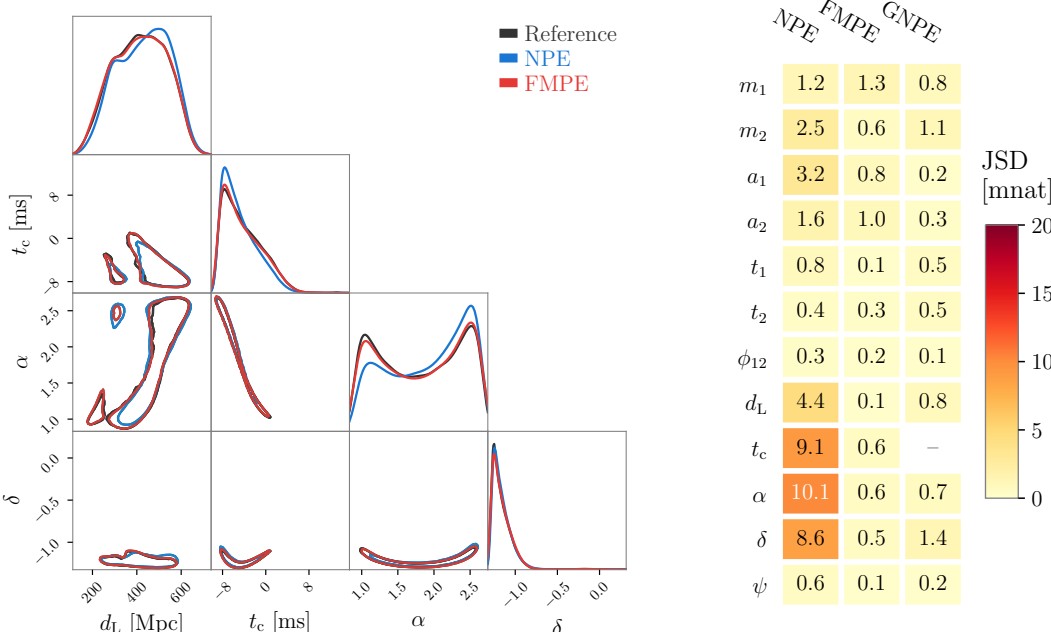

Figure 5: Results for GW150914 [72]. Left: Corner plot showing 1D marginals on the diagonal and 2D 50% credible regions. We display four GW parameters (distance $d_L$, time of arrival $t_c$, and sky coordinates $\alpha, \delta$); these represent the least accurate NPE parameters. Right: Deviation between inferred posteriors and the reference, quantified by the Jensen-Shannon divergence (JSD). The FMPE posterior matches the reference more accurately than NPE, and performs similarly to symmetry-enhanced GNPE. (We do not display GNPE results on the left due to different data conditioning settings in available networks.)

In future work we plan to carry out a more complete analysis of GW inference using FMPE. Indeed, GW150914 is a loud event with good data quality, where NPE already performs quite well. DINGO with GNPE has been validated in a variety of settings [7, 68, 40, 73] including events with a larger performance gap between NPE and GNPE [68]. Since FMPE (like NPE) does not integrate physical symmetries, it would likely need further enhancements to fully compete with GNPE. This may require a symmetry-aware architecture [74], or simply further scaling to larger networks. Nevertheless, our results demonstrate that FMPE is a promising direction for future research in this field.

## 6   Conclusions

We introduced flow matching posterior estimation, a new simulation-based inference technique based on continuous normalizing flows. In contrast to existing neural posterior estimation methods, it does not rely on restricted density estimation architectures such as discrete normalizing flows, and instead parametrizes a distribution in terms of a conditional vector field. This enables more flexible network architectures and seamless scaling (like score matching), while enabling flexible path specification and direct access to the posterior density.