# OpenReview forum: "Flow Matching for Scalable Simulation-Based Inference"
_NeurIPS.cc/2023/Conference — NeurIPS 2023 poster_

### Official Review · Reviewer_VmG3 · 2023-07-03

**Soundness:** 4 excellent
**Presentation:** 3 good
**Contribution:** 3 good
**Rating:** 7
**Confidence:** 3

**Summary:**

The paper introduces a new solution for Simulation Based Inference (SBI) by proposing Flow Matching Posterior Estimation (FMPE) as an alternative to Neural Posterior Estimation (NPE). The authors leverage the power of Flow Matching (FM) generative models as amortizors to enhance the accuracy and efficiency of SBI.
The authors emphasize that FM is a well-suited approach for SBI due to its advantageous characteristics, supported by empirical and theoretical evidence. These characteristics include easy sampling, tractable likelihood estimation, and mass coverage of the posterior distribution.
The paper details a series of experiments conducted on domain-relevant tasks to demonstrate the superiority of the FMPE method. The results showcase the effectiveness of their proposed approach compared to existing methods.

**Strengths:**

- Well written paper
- Nice theoretical results which provide good intuitions despite constrained hypotheses

**Weaknesses:**

- The likelihood of FM is very expensive (see Table 5 in App D) making it unsuited for SBI. This issue is not discussed in the paper.
- The SBI benchmark suggests good sampling capabilities but Fig 4 in Sec 4 and in Fig 9 in App C suggest that the density estimation task is poorly performed (despite mode coverage being achieved).
- The experimental results (see Fig 3 in Sec 4 for instance) don't convey a clear superiority of the proposed method over NPE.

**Questions:**

- Can we have a reference for the computation of Eq (54) in App B ? (just like (48) was derived from (49))
- As mentioned in Sec 3.1, [1] provides a MSE bound of the KL in the SDE framework. Did you try such models ?
- Can you further illustrate the superiority of your time prior ?
- Can you comment on the poor histogram fit in Fig 4 in Sec 4 and in Fig 9 in App C ? Is this mismatch critical or can it be tolerated ? Can we see the histograms for NPE ?
- How many integration steps were used in the experiments when sampling or computing the likelihood ? Was the dopri5 integrator always used ?
- The computational cost of the FM's likelihood is very prohibitive (see Table 5 in App D) which makes the comparison in Table 1 a bit unfair. Do you have a solution to address this issue ? (Are you using the Hutchinson's trace estimator ?)
- The training time difference between NPE and FMPE in GW (L294-L296) is very surprising. How do you explain it ?
- What is the design of the Normalizing Flows for NPE experiments ?

[1] Michael S Albergo, Nicholas M Boffi, and Eric Vanden-Eijnden. Stochastic interpolants: A unifying framework for flows and diffusions. arXiv preprint arXiv:2303.08797, 2023.

**Limitations:**

Not applicable.

---

> ### Author Rebuttal · Authors · 2023-08-09
>
> We thank the reviewer for the positive report and helpful comments.
>
> ### Weaknesses
>
> * *[Cost of likelihood evaluations.]* We agree that flow matching likelihood evaluations are expensive due to the many backward network passes required (taking ~6 minutes for the gravitational waves example). However, we disagree that this is generally prohibitive for SBI since likelihood evaluations are not needed for training when using the sample-conditional flow matching objective (4) (see also (7)). Moreover, for applications such as GW analysis, training and data simulation tasks are often much more costly than inference (training ~ days; inference with likelihoods ~ hours). We agree that the paper will benefit from an expanded discussion of this, so we will add a short paragraph in a new limitations section before the conclusions.
>
> * *[SBI benchmark density estimation.]* We found the histograms useful for illustrating when and how FMPE inaccuracies manifest, as well as to enable verification of mass coverage. Fig. 4, in particular, shows a failure case for the Two Moons task due to a small simulation budget $N = 10^3$, which goes away for larger $N$. For the majority of tasks, accuracy improves significantly with larger $N$, while for Lotka-Volterra, performance is always poor (as it is for NPE as well). The performance shown in the histograms is consistent with the c2st scores of Fig. 3, which we claim in general is competitive with NPE. We will add a statement to Sec. 4 on the consistency between c2st and histograms.
>
> * *[FMPE vs NPE for SBI benchmark.]* Because of the simplicity of the benchmark tasks, they are generally used by the SBI community as a sanity check rather than a discriminative test. Here we are not making a strong claim about superiority of FMPE vs NPE on the benchmark (l.232--237: "FMPE exhibits comparable performance to an NPE baseline model for most tasks and outperforms on several"). As pointed out in [1], similar statements are true for *any* SBI method (e.g., Sec. 3, #5): “no single method (NLE, NRE, NPE) outperforms another method on all tasks.” A central claim of our paper is the scalability to complicated observation spaces, which is depicted by the GW example (see Fig. 5, which demonstrates clear superiority over NPE).
>
> ### Questions
>
> * We will add a more detailed derivation of Eq. (54). The key observation is that
> $\partial_h \psi_t (\psi_{t+h})^{-1}(\theta)\rvert_{h=0} = - v_t(\theta) $  which can be obtained by applying $ \partial_h $ to $ \theta = \psi_{t+h} (\psi_{t+h})^{-1}(\theta) $ at $ h=0 $.
>
> * *[SDE models.]* We here focus on continuous flows with likelihood access, which the SDE formulation does not provide. As the GW example requires likelihoods for importance sampling, we have not applied SDE frameworks.
>
> * *[Superiority of new time prior.]* We have included in the PDF (attached to main rebuttal) an ablation of our time prior on the Two Moon tasks, which shows a significant performance improvement for the power-law prior (Sec 3.3).
>
> * *[SBI benchmark histograms.]* See above for detailed comments on Figs. 4 and 9. We used the NPE results reported in [1] in the paper, but we have now re-run these analyses ourselves to include the Two Moons NPE histogram in the PDF. The NPE histogram in this case looks better than the FMPE histogram, which is consistent with the c2st score for $N = 10^3$ (Fig. 3).
>
> * *[ODE integration.]* We always use the dopri5 solver to generate samples and likelihoods. The number of integration steps for the GW inference task is given in Table 5 in App. D (sample only: 248; sample and log probabilities: 350).
>
> * *[FM likelihood costs.]* For the results in the initial submission we computed the divergence exactly. We have now experimented also with Hutchinson’s trace estimator, and found that given a fixed number of integration steps, it provides a speed-up by a factor of $\approx$4. However, in practice, one may then need to use more integration steps to keep the error sufficiently small for downstream applications. We further note that improving the sampling times of diffusion models/flow matching is an active area of research and promising approaches have been proposed for further speed-ups [2,3].
>
> * *[Training time improvements.]* The key difference is that FMPE parametrizes a vector field, whereas NPE parametrizes a complex distribution via a parametrized mapping on the sample space (Fig. 1). When using neural spline flows [4] for NPE, a likelihood evaluation requires application of the coupling transforms (Sec. 2.1 in [4]), which involves the evaluation of splines for each flow layer as well as repeated conditioning on the context data. FMPE in contrast simply regresses on a vector field during training, which amounts to a simple neural network pass, which in turn is computationally faster. We will clarify this in the updated paper.
>
> * *[NPE architecture]* All NPE results reported in this paper use neural spline flows [4]. For the benchmark (Sec. 4), we report the results from [1] (see Sec. H.5 for their discussion on hyperparameters). For the GW example (Sec. 5), we ran NPE ourselves, using the configuration in [5] (see Sec. Supplemental Material => Training Data => Neural Network for their details).
>
> ### References
>
> [1] Lueckmann, Jan-Matthis, et al. "Benchmarking simulation-based inference." International conference on artificial intelligence and statistics. PMLR, 2021.
>
>
> [2] Lu, Cheng, et al. "Dpm-solver: A fast ode solver for diffusion probabilistic model sampling in around 10 steps." Advances in Neural Information Processing Systems 35 (2022): 5775-5787.
>
> [3] Shih, Andy, et al. "Parallel Sampling of Diffusion Models." arXiv preprint arXiv:2305.16317 (2023).
>
> [4] Durkan, Conor, et al. "Neural spline flows." Advances in neural information processing systems 32 (2019).
>
> [5] Dax, Maximilian, et al. "Real-time gravitational wave science with neural posterior estimation." Physical review letters 127.24 (2021): 241103.

---

> > ### Comment · Reviewer_VmG3 · 2023-08-18
> >
> > First of all, I want to apologize for my very late reply to your rebuttal. Your answers to our reviews were really well made. Based on your answer to my review I fully understand that the actual bottleneck is the training process (despite the simple regressive objective) and likelihood computations in the inference phase are still negligible. I truly appreciate your new plot to investigate the added value of the power low as well as the experiment with the approximate divergence for GW inference.
> > Like the other reviewers, I was also skeptical about the apparent simplicity of your experiments but I know fully understand that the toy experiments were meant to be sanity checks whereas the GW experiment is the real deal.
> > Given this excellent rebuttal and the other reviews, I am ready to increase my score to a solid 7.

---

### Official Review · Reviewer_7UFh · 2023-07-06

**Soundness:** 3 good
**Presentation:** 4 excellent
**Contribution:** 3 good
**Rating:** 6
**Confidence:** 3

**Summary:**

This paper proposes solving Bayesian inference problems where we would like to estimate & sample from p(theta | x) when p(theta) and p(x | theta) are known. This is done by randomly sampling from the joint distribution and training conditional generative models as amortized approximate posteriors.

Unlike ELBO-based training methods, the proposed approach does not ever sample from the model during training.

Experiments on some benchmark problems seem to be done, though this reviewer has no experience in these problems.

**Strengths:**

  - Well-written, motivated, and easy to understand.

  - Method is straightforward but seems to work well.

  - Theorem 1 is great.

**Weaknesses:**

  - Benchmarks do not seem to be be challenging.

**Questions:**

  - How is JSD computed? This would typically require sampling from the true posterior.

  - Are the true posteriors known for these problems?

  - The inference problems may not be challenging enough. Typically, to fit a approximate posterior, we would use the KL(p_model || p_true_posterior) or equivalently, maximizing the ELBO for some given x. This is done because this allows us to sample from the approximate posterior, which has much better properties than sampling from the prior p(theta). I wonder if the reason this is not discussed is because the problems are just not very challenging. For instance, can the proposed approach be used to train a Bayesian neural network?

  - In regards to Theorem 1, have you tried estimating how large the value of C is for the experimental problems? This grows exponentially with respect to time right?

  - In regards to Theorem 1, in the author's opinions, what is the most problematic assumption that may not hold in practice?

**Limitations:**

I do not see a discussion on limitations.

---

> ### Author Rebuttal · Authors · 2023-08-09
>
> We thank the reviewer for the positive review and helpful comments. We are happy that they liked our Theorem 1!
>
> ### Weaknesses
>
> * *[Benchmarks not challenging.]* As we argue in the response to all reviewers, the gravitational wave (GW) inference problem (Sec. 5) is in fact extremely challenging. We show that FMPE is the first generic (i.e., non symmetry-enhanced) SBI method that meets the accuracy requirements, significantly outperforming NPE. Regarding the SBI benchmark test suite (Sec. 4), we agree that this contains some simpler toy problems, but this is intended more to provide a broad set of tests relevant to SBI algorithms.
>
> ### Questions
>
> * *[How is JSD computed?]* In all cases we have access to reference posteriors: the SBI benchmark provides these in terms of MCMC samples, and for the GW problem we obtain samples using the method described in [1], which uses the exact likelihood corresponding to stationary Gaussian noise.
>
> * *[Inference problems challenging enough?]* See reply to weaknesses above.
>
> * *[Fitting an approximate posterior via ELBO?]* There are many ways to fit an approximate posterior. As you point out, one could use variational methods, which minimize KL(p_model || p_true_posterior). Alternatively, one could use NPE (which we provide as a baseline), which optimizes KL(p_true_posterior || p_model). Finally, one could use FMPE and optimize Eq. 7. All three of these methods directly fit the posterior (and will asymptotically recover the exact posterior, given enough training data and network capacity). **FMPE outperforms both of these alternatives, NPE (see Sec. 5) and ELBO maximization (see [2]).** Ref. [2] applies variational inference to the GW example, and FMPE achieves considerably more accurate results (see e.g., Fig. 2 in [2], and note that they only present results on simulations, but not on real data).
>
> * *[Sampling from the prior?]* The GW problem is indeed too challenging to be solved by sampling from the prior. In our pipeline we only ever sample from the prior to generate the training dataset. (This enables amortized inference across the entire prior.) The final model is conditional on the observational data and thereby estimates the posterior, not the prior (Eq. 7).
>
> * *[Bayesian neural networks?]* This is an interesting suggestion, and we would be curious to see whether flow matching can be applied to Bayesian neural networks. In this work, however, we specifically investigate SBI, which has very different requirements than fitting Bayesian networks. Our results therefore provide no indication on the feasibility of this.
>
> * *[Constant in Theorem 1]* Yes, the constant grows exponentially with time. This is the expected behavior (see, e.g., Groenwall’s inequality). In our setting we only consider the time interval [0,1]. We have not tried to estimate the constant for the experimental problems, which is a difficult task because this requires estimates of the derivatives of the unknown true posteriors.
>
> * *[What assumptions in Theorem 1 may not hold in practice?]* When parametrizing the learned vector field with a neural network the derivative and the second derivative will typically only be bounded by large constants $L$ and $L’$ (or unbounded when using non-differentiable activation functions like Relu). Then the bound will be weak. Also, it is difficult to find $L$ and $L’$ in practice.
>
> ### References
>
> [1] Dax, Maximilian, et al. "Neural Importance Sampling for Rapid and Reliable Gravitational-Wave Inference." Physical Review Letters 130.17 (2023): 171403.
>
> [2] Gabbard, Hunter, et al. "Bayesian parameter estimation using conditional variational autoencoders for gravitational-wave astronomy." Nature Physics 18.1 (2022): 112-117.

---

> ### Author Response · Authors · 2023-08-21
>
> We thank the reviewer once more for their positive and helpful review. We'll be happy to further clarify if there are any questions before the discussion window closes at 1pm EDT today.

---

### Official Review · Reviewer_SM25 · 2023-07-07

**Soundness:** 3 good
**Presentation:** 3 good
**Contribution:** 2 fair
**Rating:** 5
**Confidence:** 4

**Summary:**

The paper proposes a novel method for posterior estimation when samples of likelihood are given. Specifically, the authors propose the use of flow matching (FM) to train continuous normalizing flow-based (CNF-based) posterior distributions, a method they term "flow matching posterior estimation" (FMPE). The authors highlight that their proposed method is appealing for two primary reasons. Firstly, they emphasize that CNFs are more powerful than the conventional discrete normalizing flow-based (DNF-based) methods, as CNFs can utilize more flexible network architecture to represent complex densities.

Secondly, FM eliminates the need to run integrators for training CNFs, a factor often dissuading researchers from using them. To train a CNF, one must define parametric vector fields (VFs) and train the model by maximizing the likelihoods, a process that includes running integrators. However, FM differs from the conventional maximum likelihood estimation (MLE) method in that it strives to match the model's VFs to predefined target VFs, from a source noise to a target distribution. The source noise, often referred to as the initial distribution, commonly uses the standard normal distribution. The target VFs are inferred by assuming that conditional distributions from a single sample of the target to the source distribution take Gaussian forms, in a manner similar to the forward process in diffusion-based generative models. Interestingly, one can sample the target VFs at any specified time between the source and the target distribution, circumventing the need for running integrators.

In addition to the adoption of FM in posterior estimation, the paper demonstrates that under certain regularity assumptions, the flow matching loss provides an upper bound on the forward KL divergence between the target and the model posterior. The authors assert that this theoretical finding is crucial for posterior estimation as it suggests that the model posterior aims at covering the support of the target distribution rather than concentrating on higher density samples within the target's high probability region.

The authors also introduce some techniques to enhance FMPE. For instance, they propose the use of pyramid-like architectures from data to latent, with gated linear units to incorporate (latent, time) dependence, in place of the commonly chosen U-Net-based network architectures for CNFs in generative models. Additionally, they suggest an alternative t-weighting in the loss, superseding the uniform weighting of the original FM.

Lastly, the paper presents several experiments designed to demonstrate the effectiveness of the proposed method, including gravitational-wave inference tasks.

**Strengths:**

Overall I find that the writing is clear, concise, and well-structured, making it easy for readers to follow the arguments and understand the key points. The authors have succeeded in providing a fresh perspective on the topic, shedding new light on the subject matter and offering meaningful contributions to the machine learning communities.

**Weaknesses:**

From my perspective, the key innovation of this paper lies in its use of flow matching (FM) for posterior estimations. Essentially, the authors are training conditional models using FM, which marks a fresh approach in the field. While the aim of the study is clear, however, the technical challenges it faces seem minor. The authors could benefit from providing more details about these challenges to strengthen their argument and increase the depth of their research.

The choice of experiments further aggravates these issues. For instance, the use of gravitational-wave inference tasks isn't compelling enough. To better showcase the technical contributions of their work, such as the new network architectures or the t-weighting in the loss, the authors should consider more demanding tasks. These could provide a more substantial platform for discussing their technical contributions.

**Questions:**

N/A

**Limitations:**

Please refer to the comments provided in the weaknesses section.

---

> ### Author Rebuttal · Authors · 2023-08-09
>
>
> We thank the reviewer for the careful reading of our manuscript and positive comments on our writing and perspective. We fully agree with the reviewer that the main challenge was not purely technical; rather, **we combine existing methods in a new way, and show that this can solve a very hard task for which other generic SBI methods fail.** We believe that a major advantage of our approach lies in its straightforwardness, making it more accessible and easier to apply.
>
> * *[Provide more detail on technical challenges.]* Although flow matching for SBI is a natural extension, there are several caveats in applying a technique originally proposed for generative modeling. A central contribution of our paper is to point these out, analyze them in detail (theoretically and empirically), and provide improved design suggestions where necessary. This includes for example:
>
>     1. **Dimensionality of sample space.** Traditional successes of flow matching (and diffusion models) have been powered by architectures tailored for high-dimensional sample spaces. In SBI, by contrast, the observation space (or context) is typically high-dimensional, whereas the parameter space (or \(\theta\)-space) is low-dimensional. This requires a different type of architecture (Sec. 3.2).
>
>     2. **Focus on distribution vs. individual samples.** Flow matching has been shown to excel at generating high-fidelity individual samples. However, in SBI, our primary concern is to accurately capture the entire distributional shape. We demonstrate that flow matching also excels at this task.
>
>     3. **Conservative in estimates**: SBI requires conservative estimates to avoid dismissing scientifically plausible scenarios. We explored this aspect in detail from both theoretical and empirical standpoints, showing that FMPE does indeed provide probability mass covering estimates.
>
> * *[Gravitational-wave inference task demanding enough?]* GW inference is in fact an *extremely* challenging task (incl. high-dimensional observations, complicated physics models, low signal-to-noise ratio), with very high accuracy requirements (see l.301-302) and the potential for great scientific impact (also see the general reply). The only SBI method shown to meet these requirements relies on a symmetry-aware framework and thereby solves an arguably easier problem (l.276-280). FMPE is the only *generic* SBI method with comparable accuracy, and substantially outperforms other competing methods (see general reply).
>
> * *[More demanding example as platform for discussing technical contributions?]* As mentioned above, GW inference is a great platform to push the limits of SBI algorithms, and we discuss how our proposed network architecture contributes to the superior performance (l.310-317). We will expand this discussion in the updated version of the paper.

---

> ### Author Response · Authors · 2023-08-21
>
> We thank the reviewer once more for their positive and helpful review. We'll be happy to further clarify if there are any questions before the discussion window closes at 1pm EDT today.

---

### Official Review · Reviewer_1iUK · 2023-07-07

**Soundness:** 3 good
**Presentation:** 4 excellent
**Contribution:** 3 good
**Rating:** 7
**Confidence:** 4

**Summary:**

This a **very well written** paper presenting an extension to the flow matching approach for generative modelling to the simulation-based inference setting. It goes in line with preceding works that adapted normalizing flows to SBI as well.

The core idea of the paper is described in Equation (7): one can approximate (in average) the target posterior distribution by sampling $(\theta, x)$ from the joint pdf. This is rather unsurprising and it is how amortized neural posterior estimation (NPE) works. Nevertheless, this had not yet been done in the literature (at least to my knowledge) so it deserves to be shared to the community.

Furthermore, the authors have the merit of including a section on the limitations of the posterior approximation (Section 3.1) which is not often done in the SBI literature. The experiments using the `sbi-benchmark` are a very nice addition as well.

If I was going to mention one thing that bothered me about the paper was the fact of advertising their FMPE method as allowing for scalable SBI. In fact, what they present deals very well with large dimensions for the **observation space** (e.g. $x$) but the challenges related to high-dimensional **parameter space** (i.e. $\theta$) remains.

In all, I was very pleased to read this paper and it pushed me to learn some new topics that I had only heard so far but had not yet had the time to dig further (flow matching, neural ODE, continuous normalizing flows). I'm voting for a 7 (accept) and am willing to increase my score if the authors answer my questions and consider my remarks/suggestions.

**Strengths:**

- Very good writing and style of explanations.

- Simple (in the sense of not being very far reached from previous works) yet relevant new method for simulation-based inference.

- Validation experiments on the `sbi-benchmark` allowing for easy comparisons against previous works in the literature.

**Weaknesses:**

- Results on real data seem rather preliminary and just for illustrative purposes. Previous work had already shown that GNPE was good for the gravitational waves setting because of its group invariances. Why one would want to use FMPE for it then?

- Some mathematical notation are slightly inconsistent and hard to follow (I will be more precise in my questions)

- Saying that the method is **scalable** without being very precise in which sense -- in observation space, not parameter space.

**Questions:**

- Line 76: Is training time really one of the main goals when comparing methods for SBI? I would have expected more reflections around simulation budget (which, of course, is directly related to time).
- I find it very strange to refer to $\theta$ as **samples** such as in Line 100. The authors also say "sample space" when often in the SBI literature we say rather "parameter space".
- It was not super clear to me why you have to sample $\theta_1$ and what does this mean in the whole setup
- Can you give more intuition in to the failure cases for the assumptions in Theorem 1?
- By the end of your Section 3 there's no mention of how the choices of loss, architecture, and training procedure can help in improving the scalability of SBI. In fact, this feature is mainly mentioned in the title, abstract, and intro, but not that much commented in the rest of the paper.
- Figure 3: You don't give any details about how the NPE was setup in the experiments. Can you give any intuition as to why things work so badly for the Lotka-Volterra example?
- Line 296: Do you have any intuition as to why FPME trains so much faster as compared to the other methods considered in your experiments?
- Line 313: What do you mean with the much simpler vector field of FMPE? Can you give more elements into this comment?

**Limitations:**

The authors make no comment regarding the calibration and consistency of the posterior distribution obtained with their FMPE. Also, the results on the Lotka-Volterra in Figure 3 give the impression that the method can be improved for time series.

---

> ### Author Rebuttal · Authors · 2023-08-09
>
> We thank the reviewer for the thorough report and are pleased to hear that they enjoyed reading the paper.
>
> ### Weaknesses
>
> * *[“Results on real data seem rather preliminary”, "Why [...] use FMPE for it then?"]* Indeed, results are not comprehensive from a physics perspective, and we plan to carry out a more in-depth analysis in the future (l.318-321). Nevertheless, FMPE is highly promising as the first generic method to achieve the desired accuracy for GW inference (i.e., not using group equivariances) and unlike GNPE it allows for density estimation. We believe that it is reasonable to expect that once symmetries are incorporated into FMPE, it should surpass GNPE performance. This extension is planned for a follow-up study. We will update the discussion Sec. 5.3 to clarify this.
>
> * We will ensure consistency of notation in the updated version.
>
> * We indeed intended to refer to scalability in observation space, and will clarify this in the updated paper. [Note that we additionally expect FMPE to scale to high dimensional parameter spaces, as this property has been extensively demonstrated in the context of generative modeling. However, we did not show this in this paper.]
>
> ### Questions
>
> * *[Training time vs simulation budget.]* This depends on the problem at hand. For gravitational waves, our training time ($\approx$2 days on 1 GPU, 32 CPUs) is substantially larger than the simulation time ($\approx$1 hour on 64 CPUs), making the former the bottleneck. Simulation efficiency is however an important consideration in other SBI problems. We will include the simulation time in App. D.2.
>
> * *[Parameter space vs sample space.]* We agree that in the SBI context "parameter space" is the common terminology for $\theta$. We used "sample space" to be consistent with the flow matching/diffusion literature, and we will clarify this in a footnote.
>
> * *[Sampling $\theta_1$.]* $\theta_1$ refers to a parameter sample from the training dataset, sampled from the joint distribution $(\theta_1, x) \sim p(\theta_1, x)$ in the standard SBI fashion. $\theta_1$ is then used to condition the transport paths between base and target distribution, as proposed by [1]. This way, we effectively bypass the need to estimate the otherwise intractable marginal vector field between the base and the target distributions.
>
> * *[Theorem 1 failure cases.]* The assumptions fail to hold, e.g., when non-differentiable activation functions are used in the architecture.
>
> * *[Impact of choice of loss, architecture, and training procedure on scalability.]* Thank you for pointing out that we are missing this; we will add a paragraph in the discussion. Briefly, scalability is improved by regressing on a vector field rather than an entire distribution: we empirically find that fewer network parameters are needed to learn this vector field. Given the same computational budget, this allows for larger networks, ultimately enabling them to solve more complex problems (see also final response below). Furthermore, our architecture for FMPE (a straightforward ResNet with GLU conditioning) facilitates parallelization and allows for cheap forward/backward passes.
>
> * *[NPE setup for benchmark? Poor FMPE Lotka-Volterra performance?]* We did not run the NPE baseline tests ourselves, but reported the results from [2]. As pointed out in [2], no existing SBI method successfully solves Lotka-Volterra (Sec. 3, #2). Ref. [2] does not comment on the reasons for this. The Lotka-Volterra model is defined by a set of coupled differential equations (see T.10 in [2]), and presumably the pattern between model parameters and observations induced by these is hard to learn.
>
> * *[Why does FMPE train faster than other methods?]* The key difference is that FMPE parametrizes a vector field, while NPE parametrizes a complex distribution (see Fig. 1). When using neural spline flows [3] for NPE, a likelihood evaluation requires application of the coupling transforms (Sec. 2.1 in [3]), which involves the evaluation of splines for each flow layer. The flow layers are also repeatedly conditioned on the context data. FMPE in contrast simply regresses on a vector field during training, which amounts to a simple neural network pass, taking the context data as input, which is computationally faster. We will clarify this in the updated paper.
>
> * *[What is meant by "simple" vector field on l.313?]* Thank you for this question, as it forced us to think more precisely about what we mean. In fact, there are three main aspects that make the continuous NF (CNF) architecture used in FM simpler than the NF: (1) In each transform step the CNF applies a translation, whereas the NF uses an elementwise spline; (2) Although both networks use (in our case) ResNets, the NF requires a separate ResNet for each step, whereas a single `t`-dependent ResNet suffices for the CNF; and (3) The connection between network outputs and the transform is more transparent for the CNF. Although all of these contribute to "simplicity", we believe that remark (2) is key to the reduction in required (computational) complexity for good results. We will expand upon the discussion in the paper to clarify.
>
> ### Limitations
>
> We will add a short section on limitations just before the conclusions. We have now performed calibration tests (P-P plots) for the SBI benchmark task and we include representative examples in the PDF (see main rebuttal). Those results were obtained as described in  [4].
>
> ### References
>
> [1] Lipman, Yaron, et al. "Flow matching for generative modeling." arXiv preprint arXiv:2210.02747 (2022).
>
> [2] Lueckmann, Jan-Matthis, et al. "Benchmarking simulation-based inference." International conference on artificial intelligence and statistics. PMLR, 2021.
>
> [3] Durkan, Conor, et al. "Neural spline flows." Advances in neural information processing systems 32 (2019).
>
> [4] Talts, Sean, et al. “Validating Bayesian Inference Algorithms with Simulation-Based Calibration.” arXiv preprint arXiv:1804.06788 (2018)

---

> > ### Comment · Reviewer_1iUK · 2023-08-16
> > **Thank you.**
> >
> > I thank the authors for their answers and clarifications. I did enjoy reading their paper and would be pleased to see them present it in December. However, I don't see specific reasons to increase my score even further, so I will just keep it as it is.
> >
> > Cheers,
> > 1iUK

---

### Author Rebuttal · Authors · 2023-08-09

We thank the reviewers for the positive evaluation of our work and for providing detailed comments and suggestions for improvement. In particular, all reviewers recommend acceptance to varying degrees, with the paper “very well written” (1iUK), “easy to understand” (7UFh), “offering meaningful contributions to the machine learning communities” (SM25) and with “nice theoretical results” (VmG3). There is general consensus that our presentation is clear and the work relevant to the field of Simulation-Based Inference (SBI).
The most common concern raised by reviewers relates to the perceived difficulty of our experiments and whether Flow Matching Posterior Estimation (FMPE) represents a substantial improvement over the state-of-the-art in SBI. In fact, our two-pronged approach, which involves a set of benchmark tests and a real-world problem, is designed to probe complementary aspects of the method, covering breadth and depth of applications:
1. **SBI benchmark [1]:** This establishes the applicability of FMPE across a wide range of SBI challenges. Indeed, the benchmark comprises a standard set of tests for which various methods (NPE, NRE, NLE, etc.) have been optimized. The results of Sec. 4 therefore allow “for easy comparisons against previous works in the literature” (1iUK). Some of these tests do involve toy models, but they have been chosen for reasons described in Sec. 2.3 of [1], e.g., to test how performance depends on dimensionality, complexity of posteriors, and the presence of distractors. We emphasize that the benchmark does contain some challenging tasks as well (Sec. 3, #2 in [1]: “This highlights that our problems &mdash; though conceptually simple &mdash; are challenging, and there is room for development of more powerful algorithms.”). Results on these tests show that FMPE is competitive with NPE, and also serve to validate our theoretical claims of probability-mass coverage.
2. **Gravitational wave inference:** This task is designed to push the performance limits of FMPE in a real-world problem. Indeed, this is arguably one of the most challenging scientific inference tasks for which rigorous quantitative comparisons are available, allowing for precise comparisons. It involves high-dimensional data (15744-dim, see l.315), complicated physics models, low signal-to-noise ratios, multi-detector measurements and potentially non-stationary noise in the real data. Furthermore, strict accuracy requirements must be met for results to be scientifically useful (l.301-302, see also [2]). In l.276, we cite eight previous studies applying SBI to GW inference (incl. NPE, NRE, variational inference), and (unlike FMPE) no other generic method (i.e., not leveraging additional symmetry information) meets these requirements.

To summarize, we believe that our experiments do indeed represent a thorough testing ground for FMPE, and that our method displays excellent performance. To update our paper, we will add a discussion that clarifies this reasoning behind our tests. As requested by reviewers, we will also clarify what we mean by "scalability" (it refers to data dimension) and provide additional details on limitations (e.g., slower inference than NPE). All other review comments are addressed in the individual responses.

[1] Lueckmann, Jan-Matthis, et al. "Benchmarking simulation-based inference." International conference on artificial intelligence and statistics. PMLR, 2021.


[2] Romero-Shaw, Isobel M., et al. "Bayesian inference for compact binary coalescences with bilby: validation and application to the first LIGO–Virgo gravitational-wave transient catalogue." Monthly Notices of the Royal Astronomical Society 499.3 (2020): 3295-3319.

---

### Comment · Area_Chair_oqmm · 2023-08-15
**Author-reviewer discussion**

Dear all,

The author-reviewer discussion period has now started. It will continue for one more week, until August 21.

@authors: Please respond to the comments or questions reviewers may further have. Remain short and to the point.

@reviewers: Please read the author's responses and ask any further questions you may have. To facilitate the decision by the end of the process, please also acknowledge that you have read the responses and indicate whether you want to update your evaluation.

- You can update your evaluation positively (if you are satisfied with the responses) or negatively (if you are not satisfied with the responses or share other reviewers' concerns). Please note that major changes are a reason for rejection.
- You can also keep your evaluation unchanged. In this case, please indicate that you have read the responses and that you do not have any further comments.

Best regards,
The AC

---

### Decision · Program_Chairs · 2023-09-21

**Decision:**

Accept (poster)

**Comment:**

The reviewers unanimously recommend acceptance (7-5-6-7). Minor issues have been raised and discussed during the author-reviewer discussion period. The authors are encouraged to take this feedback into account when preparing the final version of the paper.